# In situ combinatorial synthesis of degradable branched lipidoids for systemic delivery of mRNA therapeutics and gene editors

Xuexiang Han [1,11,12], Junchao Xu [1,12], Ying Xu [2], Mohamad-Gabriel Alameh [3,4], Lulu Xue [1], Ningqiang Gong [1], Rakan El-Mayta [3], Rohan Palanki[1], Claude C. Warzecha [5], Gan Zhao [6], Andrew E. Vaughan [6], James M. Wilson [5], Drew Weissman[3,4] & Michael J. Mitchell [1,4,7,8,9,10] ✉

The ionizable lipidoid is a key component of lipid nanoparticles (LNPs). Degradable lipidoids containing extended alkyl branches have received tremendous attention, yet their optimization and investigation are underappreciated. Here, we devise an in situ construction method for the combinatorial synthesis of degradable branched (DB) lipidoids. We find that appending branch tails to inefficacious lipidoids via degradable linkers boosts mRNA delivery efficiency up to three orders of magnitude. Combinatorial screening and systematic investigation of two libraries of DB-lipidoids reveal important structural criteria that govern their in vivo potency. The lead DB-LNP demonstrates robust delivery of mRNA therapeutics and gene editors into the liver. In a diet-induced obese mouse model, we show that repeated administration of DB-LNP encapsulating mRNA encoding human fibroblast growth factor 21 alleviates obesity and fatty liver. Together, we offer a construction strategy for high-throughput and cost-efficient synthesis of DB-lipidoids. This study provides insights into branched lipidoids for efficient mRNA delivery.

Messenger RNA (mRNA) technology holds great promise for the treatment and prevention of a variety of pathological conditions, including cancers, infectious diseases, metabolic disorders, and congenital diseases[1,2]. Indeed, mRNA-based technology has achieved clinical success in vaccines, protein supplementation therapies, and gene editing therapies[2–5]. Specifically, two mRNA vaccines (i.e., Spikevax® and Comirnaty®) have been approved by the Food and Drug Administration (FDA) for COVID-19 prevention. mRNA is a large, negatively charged, and unstable molecule, which needs a carrier for efficient intracellular delivery[6,7].

[1]Department of Bioengineering, University of Pennsylvania, Philadelphia, PA 19104, USA. [2]Department of Chemistry, Case Western Reserve University, Cleveland, OH 44106, USA. [3]Department of Medicine, University of Pennsylvania, Philadelphia, PA 19104, USA. [4]Penn Institute for RNA Innovation, Perelman School of Medicine, University of Pennsylvania, Philadelphia, PA 19104, USA. [5]Gene Therapy Program, Perelman School of Medicine, University of Pennsylvania, Philadelphia, PA 19104, USA. [6]Department of Biomedical Sciences, School of Veterinary Medicine, University of Pennsylvania, Philadelphia, PA 19104, USA. [7]Abramson Cancer Center, Perelman School of Medicine, University of Pennsylvania, Philadelphia, PA 19104, USA. [8]Institute for Immunology, Perelman School of Medicine, University of Pennsylvania, Philadelphia, PA 19104, USA. [9]Cardiovascular Institute, Perelman School of Medicine, University of Pennsylvania, Philadelphia, PA 19104, USA. [10]Institute for Regenerative Medicine, Perelman School of Medicine, University of Pennsylvania, Philadelphia, PA 19104, USA. [11]Present address: Key Laboratory of RNA Innovation, Science and Engineering, CAS Center for Excellence in Molecular Cell Science, Shanghai Institute of Biochemistry and Cell Biology, Chinese Academy of Sciences, University of Chinese Academy of Sciences, 320 Yue Yang Road, Shanghai 200031, China. [12]These authors contributed equally: Xuexiang Han, Junchao Xu. ✉e-mail: mjmitch@seas.upenn.edu

Lipid nanoparticles (LNPs) are the most clinically advanced non-viral platforms for mRNA delivery[8–10]. LNPs are typically comprised of lipidoids (also known as ionizable lipids), phospholipids, cholesterol (Chol), and polyethylene glycol (PEG)-conjugated lipids[9,11,12]. Lipidoids play a key role in protecting and transporting mRNA cargo. At acidic pH, lipidoids are positively charged, allowing them to condense mRNA during LNP formulation and disrupt the endosomal membrane during cellular internalization. At physiological pH, lipidoids maintain a neutral charge, limiting toxicity and improving LNP pharmacokinetic properties[8]. Lipidoids typically contain an ionizable headgroup and two (or more) alkyl tails that are connected by linkers[13–15]. Notably, degradable linkers—primarily ester bonds—are preferentially used in lipidoids to allow for hydrolysis in vivo with improved biocompatibility[8,16]. In addition, branched alkyl tails are favored, since these tails can promote the lipidoid to adopt a cone-shaped structure that is beneficial for endosomal escape and mRNA delivery[17,18]. Recently, Harashima et al. conducted a mechanism-based study and further showed that branched tails could enhance the stability, ionization ability, and efficacy of LNPs[19]. Notably, two lipidoids were successfully used in the approved COVID-19 mRNA vaccines, both of which have degradable ester bonds and branched tails, further validating the merits of these structural parameters for efficacious lipidoid design[8,20–22].

However, the synthesis of degradable lipidoids with extended alkyl branches is laborious due to the lack of commercially available branched building blocks[19], which hampers their systematic optimization and investigation. In previous studies, these lipidoids were exclusively constructed based on two main steps: first, the preparation of a branched tail intermediate containing a functional group (e.g., bromo group[20,23], aldehyde group[18,22], acrylate group[24–26], hydroxyl group[27,28], and carboxyl group[19]); second, the connection of branched tail(s) to the headgroup. This construction method typically involves tedious synthesis and purification (Fig. S1), making it difficult to systemically optimize these chemical structures and investigate structure-activity relationships (SARs). We aim to develop new construction methods that enable rapid, cost-efficient, and high-throughput synthesis of degradable branched lipidoids using readily available building blocks.

Here, we devised a construction strategy for tandem and in situ combinatorial synthesis of degradable branched (DB) lipidoids based on a one-pot, two-step, three-component reaction (3-CR). In this design, two branch tails are appended to an inefficacious aminoalcohol lipidoid containing two short body tails in situ via degradable linkers, which dramatically boosts mRNA delivery (Fig. 1). We systematically synthesized and screened two combinatorial libraries of DB-lipidoids by varying the headgroup structure, length of the body tails, and length of the branch tails. Multiple DB-lipidoids were identified to form potent LNPs for in vivo mRNA delivery, which were comparable to or more efficient than the benchmark DLin-MC3-DMA (MC3) LNP formulation that was approved for hepatic delivery of small interference RNA (siRNA). Moreover, key structural criteria involving total carbon number, symmetry, and headgroup were identified, which could be used to predict the performance of unidentified DB-lipidoids. Finally, the utility of our lead DB-lipidoid was demonstrated through hepatic delivery of Cas9 mRNA/sgRNA for gene editing of transthyretin (*TTR*) and human fibroblast growth factor 21 (FGF21)-encoded mRNA for the treatment of obesity and fatty liver. With roughly five-fold greater *TTR* editing efficiency and therapeutic FGF21 protein expression than MC3 LNP, our lead DB-lipidoid—constructed via a practical synthetic method—demonstrates great promise for mRNA-based gene editing therapy and protein supplementation therapy.

## Results

### Design and construction of DB-lipidoids

A tandem and in situ construction method based on a one-pot, two-step, 3-CR was developed (Fig. 1a), which enabled rapid and parallel

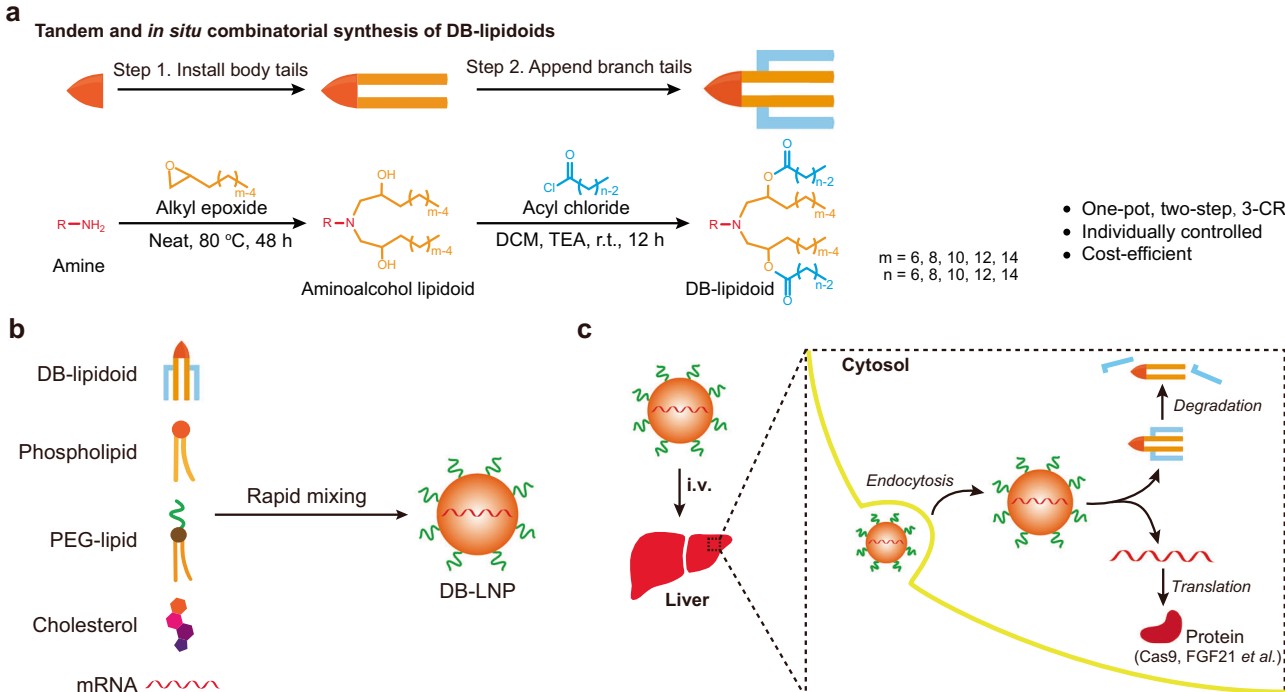

**Fig. 1 | Construction of DB-lipidoids and DB-LNP-mediated mRNA delivery. a** A scheme describing the tandem and in situ combinatorial synthesis of DB-lipidoids based on a one-pot, two-step, 3-CR. An amine reacts with alkyl epoxide (body tail) and the resulting aminoalcohol lipidoid further reacts with acyl chloride (branch tail) in situ to afford DB-lipidoid. **b** A scheme describing LNP formulation. The ethanol solution containing DB-lipidoid, phospholipid, PEG-lipid, and cholesterol is rapidly mixed with the acidic aqueous solution containing mRNA to formulate DB-LNP. **c** A scheme describing DB-LNP-mediated hepatic mRNA delivery. Intravenously (i.v.) administered DB-LNP is taken up by liver cells. mRNA is translated into protein (e.g., Cas9 and FGF21), and DB-lipidoid undergoes degradation.

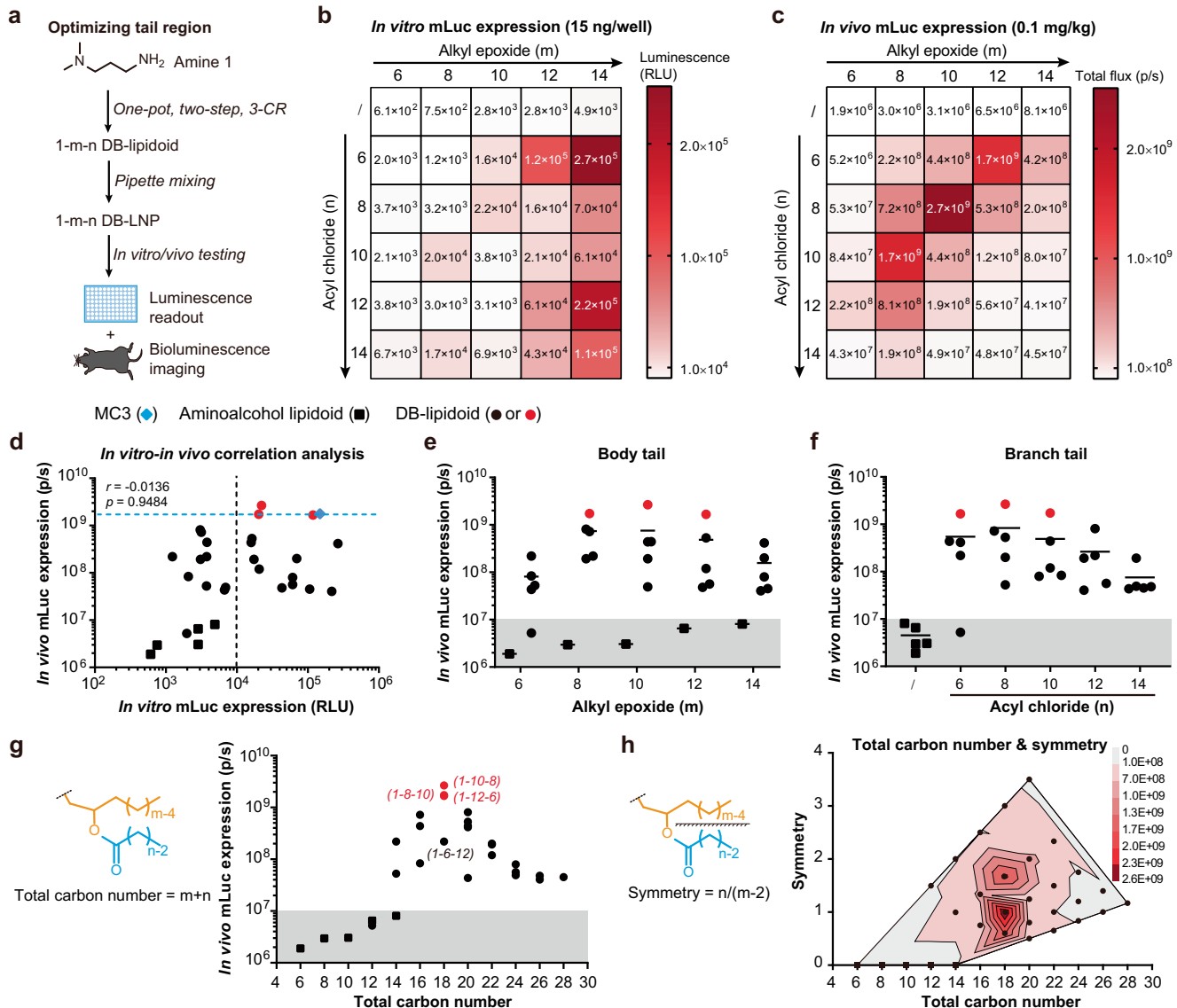

**Fig. 2 | Optimization of the tail region and screening of Library 1. a** A workflow for the synthesis and evaluation of Library 1. **b** In vitro mLuc expression shown in a heat map ($n = 3$ biologically independent samples). HepG2 cells were treated with mLuc-loaded LNPs at an mRNA dose of 15 ng/well (0.24 nM) for 24 h. RLU, relative light unit. Data are presented as mean. **c** In vivo mLuc expression shown in a heat map ($n = 3$ biologically independent samples). Mice were i.v. injected with mLuc-loaded LNPs at an mRNA dose of 0.1 mg/kg. Bioluminescence imaging (BLI) was performed at 4 h post-treatment and total flux was quantified. Data are presented as mean. **d** Correlation between in vitro and in vivo results of DB-lipidoids. The black dashed line indicates 10,000 RLU in vitro. The blue dashed line indicates the performance of MC3 LNP in vivo. Statistical significance was evaluated by a two-tailed correlation analysis using GraphPad Prism 8.0. **e** Body tail-activity relationship ($n = 3$ biologically independent samples). **f** Branch tail-activity relationship ($n = 3$ biologically independent samples). **g** Total carbon number-activity relationship. **h** Total carbon number-symmetry-activity relationship. The grey shadows indicate background levels. Source data are provided as a Source Data file.

synthesis of DB-lipidoids. In this approach, an epoxide serves as a hydroxyl group precursor: a primary amine first undergoes epoxide-mediated ring-opening reaction twice under neat conditions to afford an aminoalcohol lipidoid with two body tails, and the in situ generated hydroxyl groups further undergo acylation with acyl chlorides to attach two branch tails via ester bonds. Since the headgroup, body tail, and branch tail can be independently controlled by using a variety of inexpensive and commercially available amines, epoxides, and acyl chlorides, this 3-CR is ideal for high-throughput and cost-efficient establishment of a large and systematic combinatorial library of branched lipidoids for screening and SAR analysis.

Moreover, since i) both reactions are highly efficient with an overall yield above 80%, ii) the solvent dichloromethane (DCM) and base triethylamine (TEA) can be easily removed, and iii) the leftover reactive acyl chloride is quenched in ethanol, the crude DB-lipidoids can be directly used for LNP formulation and screening. To the best of our knowledge, this construction method represents the simplest way to produce structurally diverse degradable branched lipidoids, in comparison to previous studies involving tedious synthesis and purification (Fig. S1)[19,20,27].

**Optimizing the tail region of DB-lipidoids**

We aimed to systematically optimize the structure of DB-lipidoids and investigate SARs. Since there are three moieties (headgroup, body tail, and branch tail) that can be altered in DB-lipidoids (Fig. 1a), we decided to optimize the tail regions first and then optimize the headgroup in order to screen a large combinatorial library of DB-lipidoids in a resource-effective manner. Therefore, in the first library (Library 1), the headgroup was kept constant as amine 1 (i.e., 3-(dimethylamino)−1-propylamine) and the tail regions−both body tail and branch tail−were

varied (Fig. 2a, S2 and Table S1). Amine 1 was chosen based on the studies of MC3 lipidoid, which suggest that the dimethylamino moiety with a spacer of three methylene units is effective for siRNA delivery[29,30]. In addition, epoxides with short alkyl chains (five variations between 6 and 14 carbons) were used as body tails to minimize the molecular weights (<500 Da) of non-degradable metabolites (Fig. 1a and S2), since previous studies have suggested that small-molecule metabolites tend to undergo rapid elimination[16,20]. Correspondingly, acyl chlorides with short alkyl chains (five variations between 6 and 14 carbons) were selected as branch tails. DB-lipidoids were denoted as 1-m-n (Fig. 2a), where m and n are the number of carbons in the body tail and branch tail, respectively.

The resulting 25 DB-lipidoids in Library 1 were formulated into 25 DB-LNP formulations by pipette mixing along with 1,2-dioleoyl-sn-glycero-3-phosphoethanolamine (DOPE), Chol, and 1,2-dimyristoyl-rac-glycero-3-methoxypolyethylene glycol-2000 (DMG-PEG). The weight ratio of DB-lipidoid/DOPE/Chol/DMG-PEG was fixed at 16/10/10/3 for initial screening. To evaluate the mRNA delivery efficiency of DB-LNPs, 1-methylpseudouridine (m1ψ)-modified firefly luciferase mRNA (mLuc) was encapsulated during LNP formulation. It is should be noted that due to the divergent structures, not all lipidoids were equally suited for LNP formation and some LNP formulations might be suboptimal. In general, aminoalcohol lipidoids were inferior for mRNA encapsulation, while DB-lipidoids showed enhanced capability to encapsulate mRNA (Table S2), presumably due to the increased hydrophobicity and self-assembling ability after the attachment of two branch tails. All LNPs were between 120−230 nm in size with polydispersity index (PDI) between 0.14−0.23 and surface charge between ± 4 mV. Notably, DB-lipidoids with long body tails and branch tails (i.e., 1-12-14, 1-14-12 and 1-14-14) tended to form larger and more polydispersed DB-LNPs (Table S2), suggesting the detrimental consequence of a bulky tail region.

Afterwards, Luc expression in vitro (HepG2 cells) and in vivo after intravenous (i.v.) administration into C57BL/6 mice was determined (Fig. 2a–c). To be noted, we used low doses of mRNA for initial screening to avoid the toxicity of LNPs that could potentially affect protein synthesis (Fig. S3). For all DB-LNPs, Luc expression was mainly observed in the upper abdomen by in vivo bioluminescence imaging (BLI), which was identified to come from the liver by ex vivo BLI (Fig. S4). Interestingly, the attachment of two branch tails to aminoalcohol lipidoids boosted delivery efficiencies both in vitro and in vivo (Fig. 2b, c). Generally, the in vitro potencies of DB-lipidoids were affected by the length of the body tail (Fig. 2b), while their in vivo potencies were affected by the length of both tails (Fig. 2c). However, there was no correlation between in vitro and in vivo potencies of DB-lipidoids (Fig. 2d), a phenomenon also reported by others[31].

Next, we investigated the relationship between tail structure and in vivo potency. While all aminoalcohol lipidoids tested were inefficacious in vivo (total flux <10^7 p/s), appending branch tails dramatically boosted mRNA delivery efficiency by one to three orders of magnitude (Fig. 2e). Specifically, 8- or 10-carbon epoxide and 8-carbon acyl chloride were most effective (Fig. 2e, f), while further modulation of tail length reduced the overall potency benefit. Notably, the top three DB-lipidoids (1-8-10, 1-10-8, 1-12-6) in this library exhibited comparable potency to MC3, all of which bear a total of 18 carbons in each tail (m + n, Fig. 2d, g). Further increase or decrease in total carbon number of body and branch tails generally led to the loss of activity (Fig. 2g). We termed this phenomenon as the "18-Carbon Rule." Interestingly, we found this "Rule" to be necessary but not adequate to afford potent DB-lipidoids, as the 1-6-12 DB-lipidoid displayed suboptimal performance even though it contained a total of 18 carbons in each tail (Fig. 2g).

Recently, Harashima et al. suggested that the symmetry of a branched tail could impact potency of the lipidoid[19]. Therefore, we prepared a contour plot with both parameters−total carbon number

and tail symmetry−to better visualize their influence on in vivo performance (Fig. 2h). As expected, the potency of DB-lipidoids was dependent not only on total carbon number, but also on the symmetry of body to branch tails. The symmetry (calculated as n/(m-2)) of the lead 1-10-8 DB-lipidoid was defined as 1 due to the relatively symmetrical structure of the body and branch tails, while the symmetries of the slightly less potent 1-8-10 and 1-12-6 DB-lipidoids as well as the significantly less potent 1-6-12 DB-lipidoid were determined to be 1.7, 0.6, and 3, respectively. This finding suggests that, for DB-lipidoids with a total carbon number of 18, less symmetry (i.e., more deviation from 1) is associated with less in vivo mRNA delivery efficiency. Taken together, these results indicate that the optimal tail region should follow the "18-Carbon Rule" and have a symmetry of 1, which leads to the combination of a 10-carbon body tail and an 8-carbon branch tail.

Next, we tested the in vivo potency of purified DB-lipidoid containing two branch tails and compared it to DB-lipidoid containing one branch tail (an intermediate metabolite of the standard DB-lipidoid, Fig. S5). Purified 1-10-8 with two branch tails (denoted as 1-10-8(2)) demonstrated comparable potency to the crude product, confirming this to be the active transfection agent. In contrast, synthesized 1-10-8 with one branch tail (denoted as 1-10-8(1)) was unable to encapsulate or deliver mRNA. We then assessed the degradability of DB-lipidoids. After treatment of 1-10-8(2) with esterase (an enzyme that hydrolyzes esters) for 1 h, its metabolite 1-10-8(1) was detected (Fig. S6). Together, these results suggest that both branch tails are required for the potency of DB-lipidoid and they can be detached following degradation.

## Optimizing the headgroup of DB-lipidoids

In the second stage of our optimization scheme, we kept the tail region constant at the optimal parameters identified above and varied the headgroup to generate a second library of DB-lipidoids (i.e., Library 2, Fig. 3a, S7 and Table S1). To maintain their structural similarity, only amines that can attach two body tails were chosen. In total 20 chemically diverse amines were tested, including monoamines, diamines, polyamines, and hydrazines (Fig. S7). The majority of these amines were selected from previous publications[32,33]. DB-lipidoids were denoted as x-10-8, where x stands for amine number identifier. The in vitro and in vivo performance of DB-lipidoids in Library 2 was determined using the same experimental conditions as Library 1 (Fig. 3b, c and S8). Again, these DB-lipidoids demonstrated a poor correlation between their in vitro and in vivo potencies (Fig. 3d). In this library, six DB-lipidoids−containing headgroup 1, 7, 9, 10, 11, or 12− exhibited comparable or superior in vivo performance compared to MC3, despite that none of them outperformed MC3 in vitro (Fig. 3b-d). Interestingly, the results from Libraries 1 and 2 demonstrate that although potent in vivo DB-lipidoids do not necessarily perform well in vitro, all of them surpass a baseline mRNA transfection threshold (i.e., 10,000 RLU, Figs. 2d and 3d).

We further investigated the relationship between headgroup structure and in vivo potency. Amines with only one primary amine (2, 3, and 6), two secondary amines (16−20) or a hydrazine group (4 and 5) were unable to generate potent DB-lipidoids (Fig. 3c). Efficacious amines (1, 7, 9, 10, 11, and 12) shared certain structural similarities: diamines with one primary amine and one tertiary amine spaced by two or three carbons. Notably, both the length of spacer (e.g., 1, 7, and 8) and the form of tertiary amine (e.g., 9, 11, and 14) critically impacted in vivo performance. The optimal spacer length was identified to be two or three methylene units, while the ideal tertiary amine structure was determined to be dimethylamino, diethylamino or pyrrolidinyl (Fig. 3e).

Together, after two rounds of optimization, several structural criteria of potent DB-lipidoids were identified: (1) total carbon number = 18; (2) symmetry = 1; (3) diamines with one primary amine and one dimethylamino-, diethylamino- or pyrrolidinyl-based tertiary amine

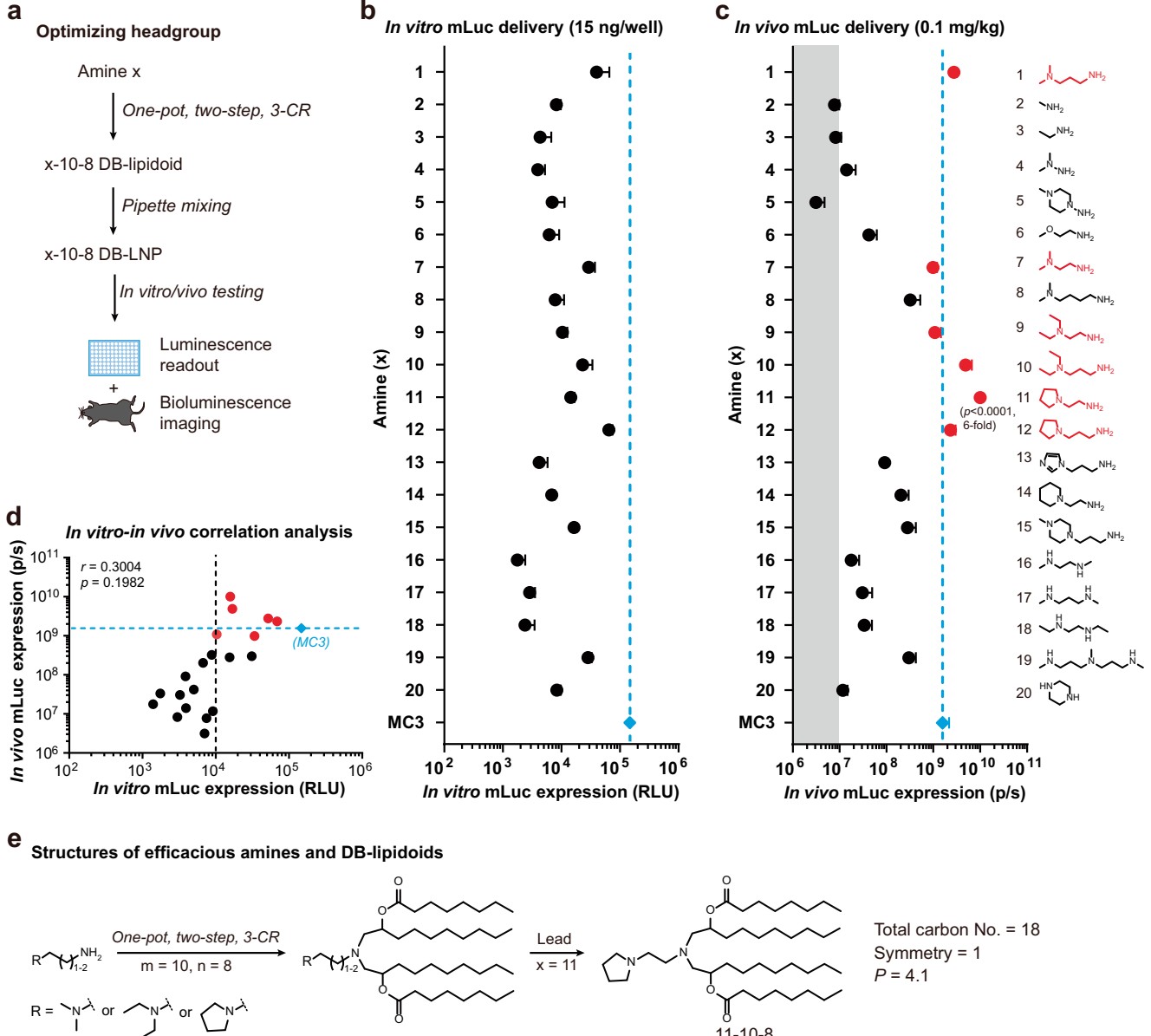

**Fig. 3 | Optimization of the headgroup and screening of Library 2. a** A workflow for the synthesis and evaluation of Library 2. **b** In vitro mLuc expression ($n = 3$ biologically independent samples). HepG2 cells were treated with mLuc-loaded LNPs at an mRNA dose of 15 ng/well (0.24 nM) for 24 h. Data are presented as mean ± SD. **c** In vivo mLuc expression ($n = 3$ biologically independent samples). Mice were i.v. injected with mLuc-loaded LNPs at an mRNA dose of 0.1 mg/kg. BLI was performed at 4 h post-treatment and total flux was quantified. Efficacious DB-lipidoids and their amines are highlighted in red. The grey shadow indicates background level. Data are presented as mean ± SD. Statistical significance was evaluated by a one-way ANOVA with Tukey's correction. **d** Correlation between in vitro and in vivo results of DB-lipidoids. The black dashed line indicates 10,000 RLU in vitro. The blue dashed line indicates the performance of MC3 LNP in vivo. Statistical significance was evaluated by a two-tailed correlation analysis using GraphPad Prism 8.0. **e** Structures of efficacious amines and DB-lipidoids. The chemical structure of lead DB-lipidoid 11-10-8 is shown. 11-10-8 demonstrates a total carbon number of 18, a symmetry of 1, and a packing parameter (P) of 4.1. Source data are provided as a Source Data file.

spaced by two or three carbons. The lead DB-lipidoid 11-10-8, roughly six-fold more potent than MC3, was identified and used for subsequent studies. Benefitting from two branch tails, the packing parameter (P) of this molecule was determined to be 4.1 according to molecular dynamics simulations (Fig. S9), suggesting that 11-10-8 adopts a more cone-shaped structure than MC3 (4.1 versus 2.11)[34].

**Predicting performance and optimizing formulation of DB-LNPs**
To further validate our structural criteria for DB-lipidoids, we sought to predict the in vivo performance of unidentified DB-lipidoids. As a case study, we synthesized four amine 11-based DB-lipidoids that either violated the "18-Carbon Rule" or defied the optimal symmetry of the tail region: 11-6-12 (total carbon number = 18, symmetry = 3), 11-8-10 (total carbon number = 18, symmetry = 1.7), 11-12-6 (total carbon number = 18, symmetry = 0.6) and 11-12-10 (total carbon number = 22, symmetry = 1). We predicted that these DB-lipidoids would have less in vivo mRNA transfection potency relative to 11-10-8. Indeed, our experimental results confirmed our prediction (Fig. 4a); all four DB-lipidoids that breached the aforementioned design criteria were less potent than 11-10-8. Moreover, despite having the same total carbon

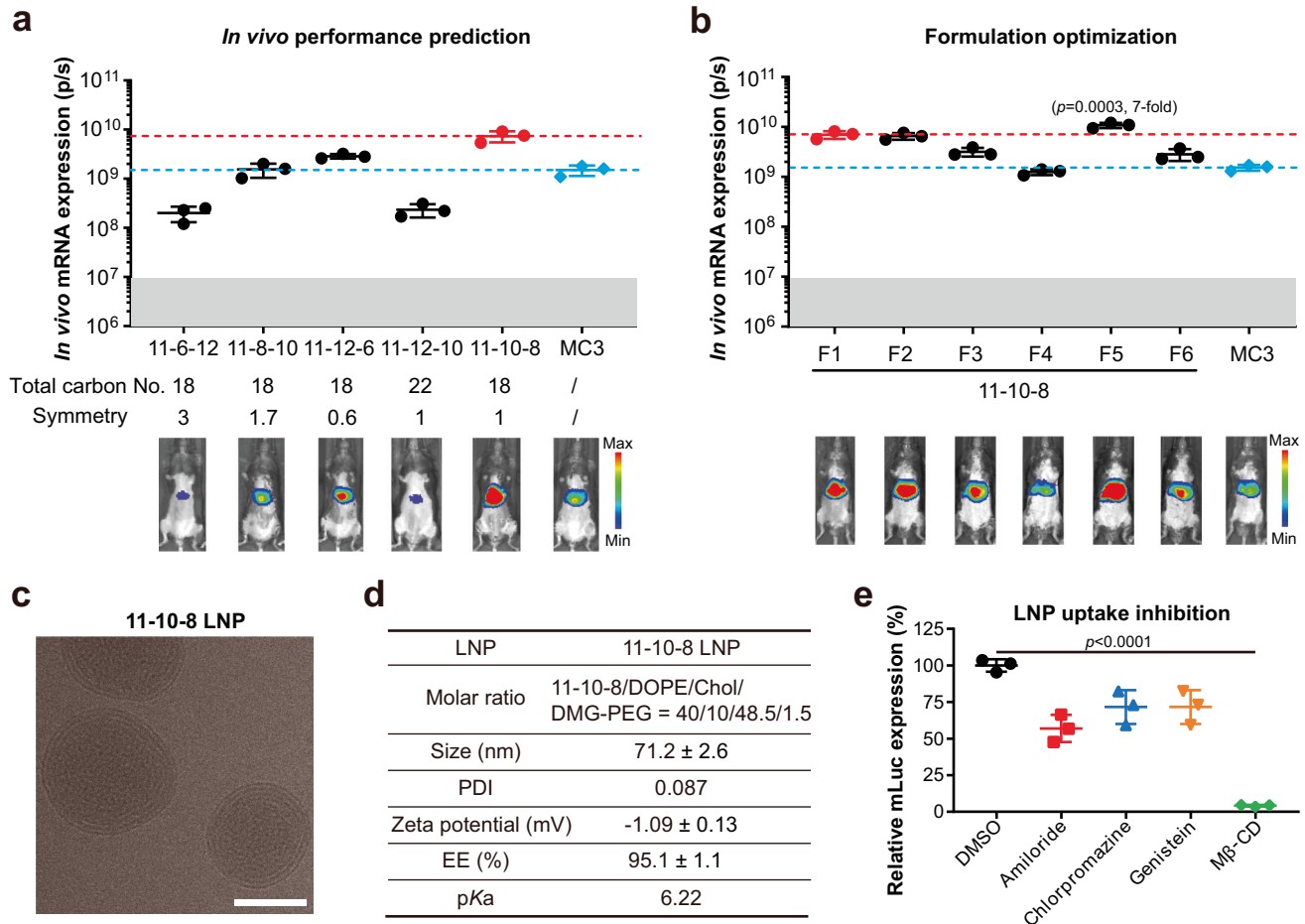

**Fig. 4 | Prediction, optimization, and characterization of DB-LNPs. a** Prediction and verification of in vivo performance for unidentified DB-lipidoids (*n* = 3 biologically independent samples). Mice were i.v. injected with mLuc-loaded LNPs at an mRNA dose of 0.1 mg/kg. BLI was performed at 4 h post-treatment and total flux was quantified. The red dashed line indicates the performance of 11-10-8 LNP, while the blue dashed line indicates the performance of MC3 LNP. The grey shadow indicates background level. **b** Optimization of 11-10-8 LNP formulation (*n* = 3 biologically independent samples). Mice were injected i.v. with mLuc-loaded LNPs at an mRNA dose of 0.1 mg/kg. BLI was performed at 4 h post-treatment and total flux was quantified. The grey shadow indicates background level. Statistical significance was evaluated by a one-way ANOVA with Tukey's correction. **c** A representative cryo-EM image of 11-10-8 LNP from three independent experiments. Scale bar = 50 nm. **d** Physicochemical properties of 11-10-8 LNP (*n* = 3). **e** Inhibition of 11-10-8 LNP uptake by various endocytic inhibitors (*n* = 3 biologically independent samples). Amiloride is an inhibitor of macropinocytosis; Chlorpromazine is an inhibitor of clathrin-mediated endocytosis; Genistein is an inhibitor of caveolae-mediated endocytosis; Methyl-β-cyclodextrin (Mβ-CD) is an inhibitor of lipid raft-mediated endocytosis. Statistical significance was evaluated by a one-way ANOVA with Tukey's correction. Data are presented as mean ± SD. Source data are provided as a Source Data file.

number of 18, 11-8-10 and 11-12-6 with more symmetry exhibited similar or higher potency compared to MC3, but 11-6-12 with less symmetry dramatically lost potency as expected. Therefore, our structural criteria can be used to predict the in vivo performance of DB-lipidoids, which could be useful for the future design and discovery of potent branched lipidoids.

Next, we optimized the formula of 11-10-8 LNP and screened a series of well-established LNP formulations (F1-6, Fig. 4b and Table S3)[29,35,36]. F1 was the formulation used in initial screening, which comprised crude 11-10-8, while purified 11-10-8 was used in F2-6 (Figs. S10 and S11). Crude 11-10-8 and purified 11-10-8 had similar in vivo potency when the same LNP formula was used (F1 versus F2, Fig. 4b), which was in line with the previous observation (Fig. S5d). Moreover, all 11-10-8 LNP formulations outperformed the benchmark MC3 LNP formulation except F4, which possessed a higher molar percentage of DMG-PEG compared to other formulations (Fig. 4b and Table S3). The top-performing formulation F5 (11-10-8/DOPE/Chol/DMG-PEG at a molar ratio of 40/10/48.5/1.5), roughly seven-fold more potent than MC3, was chosen for further studies.

## Characterization and in vitro studies of DB-LNPs

We next characterized the physicochemical properties of this optimized 11-10-8 LNP. Cryogenic electron microscopy (Cryo-EM) showed that 11-10-8 LNP possessed a dense spherical structure with a multi-lamellar shell and an amorphous core (Fig. 4c). The mRNA encapsulation efficiency (EE) of 11-10-8 LNP was determined to be ~95% (Fig. 4d). The hydrodynamic size of 11-10-8 LNP was approximately 71.2 nm with a low polydispersity index (PDI = 0.087) and a neutral surface charge (ζ = −1.09 mV). The apparent p$K_a$ of 11-10-8 LNP was determined to be 6.22 (Fig. S12). Due to its ionization ability, 11-10-8 LNP induced minimal hemolysis at pH 7.4, but increased hemolysis at pH 6.0 (Fig. S13). This ionization behavior is critical for reducing the toxicity of LNPs and enhancing membrane disruption in the acidic endosome[8,37].

In HepG2 cells, 11-10-8 LNP showed dose-dependent mLuc delivery with minimal toxicity at doses ranging from 0 to 240 ng/well (Fig. S14). Moreover, 11-10-8 LNP exhibited dose-dependent delivery of GFP mRNA (Fig. S15), and could achieve nearly 100% transfection efficiency at a dose as low as 150 ng/mL. Next, we examined the cellular

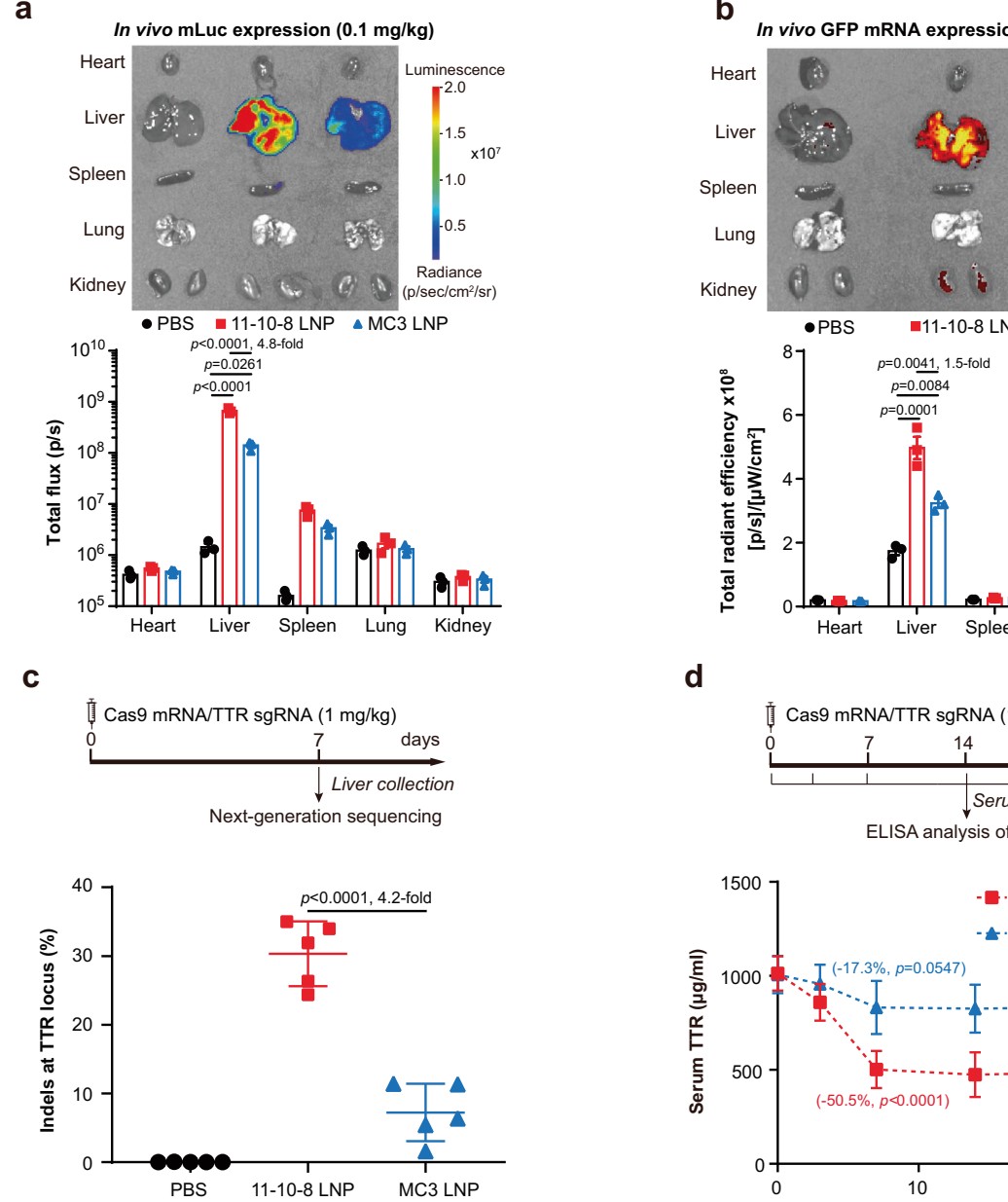

**Fig. 5 | DB-LNP-mediated hepatic delivery of mRNA-based gene editors. a** Ex vivo BLI of major organs from treated mice and their quantification ($n = 3$ biologically independent samples). Mice were i.v. injected with mLuc-loaded LNPs at an mRNA dose of 0.1 mg/kg. Images were taken at 4 h post-treatment. Statistical significance was evaluated by a one-way ANOVA with Tukey's correction. **b** Ex vivo fluorescence imaging of major organs from treated mice and their quantification ($n = 3$ biologically independent samples). Mice were i.v. injected with GFP mRNA-loaded LNPs at an mRNA dose of 0.25 mg/kg. Images were taken at 4 h post-treatment. Statistical significance was evaluated by a one-way ANOVA with Tukey's correction. **c, d** LNP-mediated Cas9 mRNA/TTR sgRNA co-delivery and gene editing. Mice were i.v. injected with LNPs co-delivering Cas9 mRNA/TTR sgRNA (4:1, wt:wt) at a total RNA dose of 1 mg/kg. Mice were euthanized on day 7, and DNA was extracted from the liver to determine on-target indel frequency by next-generation sequencing (**c**, $n = 5$ biologically independent samples). Statistical significance was evaluated by a one-way ANOVA with Tukey's correction. Serum was collected at the indicated time points for ELISA analysis of TTR (**d**, $n = 5$ biologically independent samples). Statistical significance was evaluated by a one-way ANOVA with Tukey's correction. Data are presented as mean ± SD. Source data are provided as a Source Data file.

uptake routes of 11-10-8 LNP. The endocytosis of 11-10-8 LNP was highly dependent on lipid rafts (Fig. 4e), as pre-treatment with methyl-β cyclodextrin (Mβ-CD), an inhibitor of lipid raft-mediated endocytosis, completely suppressed mLuc delivery. Other internalization pathways (e.g., macropinocytosis, clathrin-mediated endocytosis and caveolae-mediated endocytosis) were also involved but contributed less to overall LNP internalization, since their inhibitors only slightly suppressed mLuc delivery (Fig. 4e).

## Hepatic delivery of mRNA-based gene editors using DB-LNPs

Hepatic delivery of mRNA-based gene editors holds great promise for genome editing therapies. Therefore, we first demonstrated the potential of DB-LNPs for this application by comparing 11-10-8 LNP with MC3 LNP (Table S4). Ex vivo BLI results confirmed that 11-10-8 LNP predominantly transfected the liver, resulting in 4.8-fold higher Luc expression in the liver than MC3 LNP (Fig. 5a). These results were consistent with previous in vivo BLI results (Fig. 4b). Similarly, 11-10-8

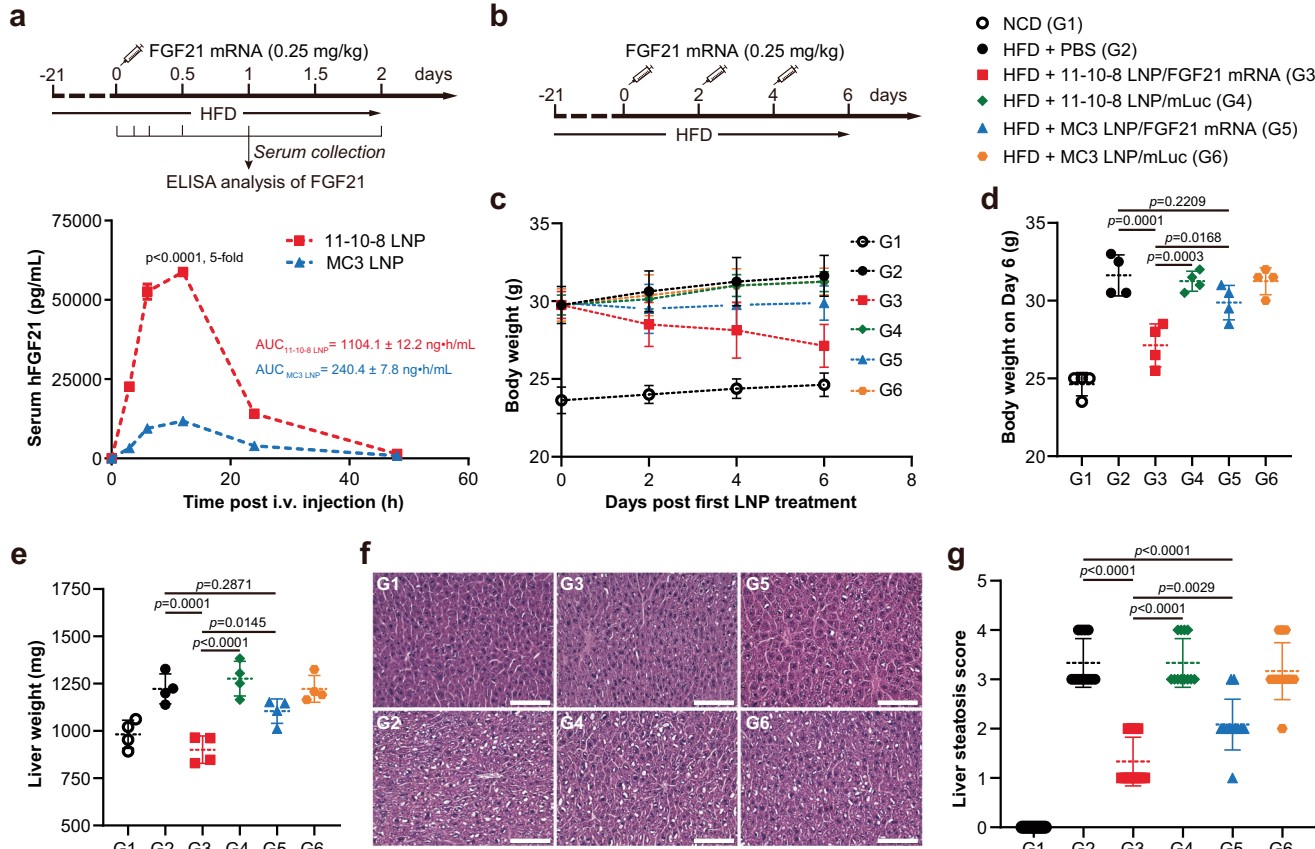

**Fig. 6 | DB-LNP-mediated hepatic delivery of FGF21 mRNA. a** LNP-mediated FGF21 mRNA delivery ($n = 4$ biologically independent samples). Male mice were fed a HFD for three weeks to induce obesity (body weight ~30 g) and fatty liver. These obese mice were i.v. injected with FGF21 mRNA-loaded LNPs at an mRNA dose of 0.25 mg/kg. Serum was collected at the indicated time points for ELISA analysis of FGF21. AUC of FGF21 exposure during the time interval 0–48 h was determined. Statistical significance was evaluated by an unpaired two-tailed Student's t-test. **b** A scheme of FGF21 mRNA therapy in HFD-induced obese mice. Obese mice were i.v. injected with various LNP formulations at an mRNA dose of 0.25 mg/kg every other day for three doses. Male mice fed a normal chow diet (NCD) were used as a control group. **c** Body weight growth curve ($n = 4$ biologically independent samples). **d** Body weight on Day 6 ($n = 4$ biologically independent samples). Statistical significance was evaluated by a one-way ANOVA with Tukey's correction. **e** Liver weight ($n = 4$ biologically independent samples). Statistical significance was evaluated by a one-way ANOVA with Tukey's correction. **f** Representative images of liver histological examinations (H&E staining). Scale bars: 100 μm. **g** Liver steatosis score ($n = 12$). A total of 12 images for each group (three random fields for each liver section, four mice per group) were analyzed semi-quantitatively for liver steatosis. Statistical significance was evaluated by a one-way ANOVA with Tukey's correction. Data are presented as mean ± SD. Source data are provided as a Source Data file.

LNP also outperformed MC3 LNP in terms of hepatic delivery of GFP mRNA (Fig. 5b). Immunofluorescence staining results further confirmed that 11-10-8 LNP mediated greater GFP expression than MC3 LNP in major liver cells, including hepatocytes, Kupffer cells and liver sinusoidal endothelial cells (Fig. S16).

Given the growing interest in CRISPR/Cas9-mediated liver gene editing as exemplified by Intellia's LNP-mediated transthyretin (*TTR*) knockout[38], we chose *TTR* as a model target and co-delivered Cas9 mRNA and TTR sgRNA. A single injection of 11-10-8 LNP co-delivering these two components at a clinically relevant dose (1 mg RNA/kg) led to ~30% insertions and deletions (indels) at the *TTR* locus and ~50% reduction of serum TTR (Fig. 5c, d). In contrast, MC3 LNP only achieved ~7% indels and ~17% reduction of serum TTR. Notably, no observable hepatotoxicity was induced by both LNP treatments as demonstrated by normal levels of alanine transaminase (ALT) and aspartate aminotransferase (AST) in treated mice (Fig. S17). Moreover, no abnormal innate immune responses were observed after examining 13 cytokines (Table S5). Together, these results strongly demonstrate the potential of our DB-LNPs in the hepatic delivery of gene editors.

## Hepatic delivery of mRNA-based therapeutics using DB-LNPs
Hepatic delivery of mRNA-based therapeutics holds great promise for protein supplementation therapies. FGF21 is a pleiotropic metabolic

hormone primarily secreted by the liver, which is a promising therapeutic agent for obesity, type 2 diabetes and non-alcoholic steatohepatitis[39,40]. We next evaluated 11-10-8 LNP for the delivery of human FGF21-encoded mRNA in high fat diet (HFD)-induced obese mice. Male mice were fed a HFD for three weeks to induce obesity (body weight ~30 g) and fatty liver[41], and then i.v. injected with FGF21 mRNA-loaded LNPs. The expression of FGF21 peaked (58.8 ± 1.1 ng/mL) at 12 h post-administration of FGF21 mRNA-loaded 11-10-8 LNP, which was five-fold higher than that in FGF21 mRNA-loaded MC3 LNP-treated mice (11.8 ± 1.1 ng/mL, Fig. 6a). Moreover, the area under curve (AUC)— a pharmacokinetic metric of therapeutic exposure—of FGF21 was 4.6-fold greater in 11-10-8 LNP-treated mice than that in MC3 LNP-treated mice (1104.1 ± 12.2 ng·h/mL *versus* 240.4 ± 7.8 ng·h/mL, Fig. 6a).

To further demonstrate the therapeutic potential of this FGF21 mRNA therapy, we examined its weight-reducing and lipid-lowering effects in obese mice[39]. Obese mice were treated with various LNP formulations every other day for three doses (Fig. 6b). While obese mice treated with PBS or mLuc-loaded LNPs gradually increased weight, obese mice treated with FGF21 mRNA-loaded 11-10-8 LNP or FGF21 mRNA-loaded MC3 LNP lost weight or maintained weight, respectively (Fig. 6c). At the end of this experiment, body weight as well as liver weight of obese mice treated with FGF21 mRNA-loaded 11-10-8 LNP was significantly reduced compared to that of obese mice treated

with other formulations (Fig. 6d, e). Hematoxylin and eosin (H&E) staining of livers showed less vacuoles and reduced liver steatosis in the obese mice treated with FGF21 mRNA-loaded 11-10-8 LNP compared to those obese mice treated with other formulations (Fig. 6f, g). It is worth mentioning that although MC3 LNP-based FGF21 mRNA therapy exhibited weight-reducing and lipid-lowering effects to some extent in obese mice, they were less obvious than 11-10-8 LNP-based therapy, presumably due to less FGF21 expression (Fig. 6a). Together, these results suggest that 11-10-8 LNP-based FGF21 mRNA therapy could be a promising approach for treating obesity and fatty liver.

While 11-10-8 LNP was more effective at delivering large RNA constructs compared to MC3 LNP, we then assessed whether it still holds advantages in delivering siRNA, for which the MC3 LNP is approved[42]. Interestingly, when TTR siRNA was delivered, 11-10-8 LNP and MC3 LNP showed comparable potency with a similar median effective dose (ED50, 0.029 mg/kg vs 0.030 mg/kg, Fig. S18a). Owing to the potent silencing effect, a single injection of 11-10-8 LNP at a clinically relevant dose (1 mg siRNA/kg) resulted in ~100% reduction of serum TTR on day 3 and nearly 40% reduction on day 28 (Fig. S18b). These results suggest that our DB-LNPs could also be a robust platform for the delivery of siRNA.

Apart from systemic delivery, we further showed that 11-10-8 LNP mediated strong intramuscular (i.m.) mRNA delivery (Fig. S19), which outperformed the benchmark SM-102 LNP that has been approved by the FDA for mRNA vaccine delivery[8]. These results demonstrate the potential of 11-10-8 LNP for local mRNA delivery and mRNA vaccine development.

## Discussion

Degradable lipidoids comprising extended alkyl branches have received tremendous interest due to their success in the clinic. However, it is challenging to build a large and systematically-designed library of branched lipidoids with varying lengths of body tail and branch tail based on previous synthetic methods (Fig. S1), making their optimization and investigation largely hampered. For the first time, we devised a tandem and in situ construction method for the rapid, cost-efficient, and high-throughput synthesis of degradable branched lipidoids (Fig. 1a). This facile construction method avoids the use of branched intermediates and allows for independent control of each structural parameter, including headgroup, body tail, branch tail, and symmetry.

We demonstrated the utility of this construction strategy through the generation of two combinatorial libraries of DB-lipidoids with varying headgroups, body tails, and branch tails (Figs. 2 and 3). Remarkably, appending two branch tails to aminoalcohol lipidoids through ester linkers boosts in vivo mRNA delivery efficiency by one to three orders of magnitude (Fig. 2e). Importantly, these branch tails can be detached following degradation (Fig. S6). Notably, we used a two-step combinatorial optimization and screening strategy involving two small libraries instead of one combinatorial screening strategy involving a large library (5 × 5 × 20 = 500 DB-lipidoids) to dramatically reduce the workload and usage of mice. In total, eight DB-lipidoids from two libraries with potency comparable to or greater than MC3 were identified (hit rate = 18%). Given the amount of time, effort and resources put into the development of MC3 and its analogs[16,29,30], such a high hit rate for our easily synthesized DB-lipidoids is appreciated.

Moreover, our study revealed key structural criteria governing DB-lipidoids potency: (1) total carbon number = 18; (2) symmetry = 1; (3) diamines with one primary amine and one dimethylamino-, diethylamino- or pyrrolidinyl-based tertiary amine spaced by two or three carbons. Importantly, these structural criteria can be used to predict the performance of unidentified DB-lipidoids and guide the discovery of potent ones (Fig. 4a). It is worth mentioning that since the total carbon number in the tail region is closely related to the

hydrophobicity of DB-lipidoids, it is reasonable that the optimal one was determined to be 18, considering that many natural and synthetic lipids contain 18-carbon tails[13]. Moreover, we found that tail symmetry could contribute to the potency of branched lipidoids, which is consistent with a recent publication[19]. However, compared to their synthetic strategy (Fig. S1), our construction method is more concise and flexible, enabling the generation of more structurally diverse branched lipidoids.

To demonstrate the potential of DB-lipidoids in mRNA-based genome editing therapy and protein supplementation therapy, we performed head-to-head comparisons against the approved, liver-tropic MC3 LNP. Remarkably, our 11-10-8 LNP enabled roughly five-fold higher *TTR* genome editing efficiency and therapeutic FGF21 protein expression compared to MC3 LNP (Figs. 5c and 6a). Moreover, 11-10-8 LNP-based FGF21 mRNA therapy exhibited superior weight-reducing and lipid-lowering effects compared to the MC3 LNP-based mRNA therapy, resulting in a significant alleviation of obesity and fatty liver in a diet-induced obese mouse model (Fig. 6b–g). Interestingly, although 11-10-8 LNP was more potent than MC3 LNP at delivering mRNA, their potency for siRNA delivery was comparable (Fig. S18). Similar results were observed by others[35], where an optimized LNP formulation could lead to a seven-fold increase in mRNA transfection, but did not enhance siRNA transfection. It is speculated that siRNA is more tolerant than mRNA for the potency of LNPs[35]. Nevertheless, the easy and cost-efficient synthesis of 11-10-8 remains advantageous for siRNA delivery compared to MC3.

There are some limitations to this study. First, to ease SAR analysis of tail regions, only saturated and linear epoxides and acyl chlorides were used in Library 1. It is worth mentioning that it is feasible to use unsaturated or branched epoxides and acyl chlorides based on our construction method, which can further increase the structural diversity of DB-lipidoids. Second, to simplify SAR analysis of head-groups, only amines that can attach two body tails were selected for Library 2, while complex amines (e.g., polyamines and dendrimers) that can attach multiple body tails were excluded. Third, while potent DB-LNPs show superior in vivo mRNA delivery compared to MC3 LNP, their in vitro performances are inferior. The reason for this discrepancy between in vitro and in vivo results is unclear at the current stage. Nevertheless, further exploration of branched lipidoids and understanding of their in vivo potency are on-going.

In conclusion, we devised a construction method that enables one-pot, high-throughput, and cost-efficient synthesis of DB-lipidoids. We identified multiple potent DB-lipidoids through combinatorial synthesis and screening of two libraries, and summarized key structural criteria governing the potency that can be used to predict the performance of unidentified analogs and guide the discovery of potent ones. Our lead DB-lipidoid outperformed the benchmark lipid MC3 in terms of hepatic mRNA delivery, demonstrating great potential for mRNA-based protein supplementation therapy and gene editing therapy. Overall, our construction method lowers the threshold for synthesizing branched lipidoids, and this study lays a foundation for the further development and application of branched lipidoids for mRNA delivery.

## Methods
### Materials
Amines, epoxides, acyl chlorides, TEA were purchased from Sigma Aldrich (Burlington, Massachusetts, USA), Tokyo Chemical Industry (TCI, Tokyo, Japan) and Ambeed (Arlington Heights, Illinois, USA). Alanine transaminase (ALT) colorimetric activity assay kit (#700260) and aspartate aminotransferase (AST) colorimetric activity assay kit (#701640) were purchased from Cayman Chemical (Ann Arbor, Michigan, USA). 1,2-dioleoyl-sn-glycero-3-phosphoethanolamine (DOPE), 1,2-distearoyl-sn-glycero-3-phosphocholine (DSPC), 1,2-dimyristoyl-rac-glycero-3-methoxypolyethylene glycol-2000 (DMG-

PEG 2000) and cholesterol were obtained from Avanti Polar Lipids (Alabaster, Alabama, USA). DLin-MC3-DMA and SM-102 were purchased from MedChem Express (Monmouth Junction, New Jersey, USA). Cas9 mRNA (5moU) was purchased from TriLink (San Diego, California, USA). Highly modified sgRNA target mouse *TTR* (guide No. G211) was chemically synthesized by AxoLabs (Kulmbach, Bayern, Germany) based on the previous publication[43]. Anhydrous DCM, porcine liver esterase, amiloride hydrochloride, chlorpromazine, genistein, methyl-beta-cyclodextrin and TTR siRNAs (#NM_013697, siRNA IDs: SASI_Mm01_00076059, SASI_Mm01_00076060 and SASI_Mm01_00076061) were purchased from Sigma Aldrich (Burlington, Massachusetts, USA).

## mRNA Synthesis
Codon optimized coding sequence of firefly luciferase, GFP sequence, or human FGF21 was cloned into an mRNA production plasmid (optimized 3′ and 5′ UTR with a 101 polyA tail)[44], in vitro transcribed in the presence of 1-methyl pseudouridine modified nucleoside, co-transcriptionally capped using the CleanCap™ technology (#N-7113, TriLink) and cellulose purified to remove double-stranded RNAs[45]. Purified mRNA was ethanol precipitated, washed, re-suspended in nuclease-free water, and subjected to quality control. All mRNAs were stored at −20 °C until use. These mRNAs are available upon reasonable request.

## General method for the synthesis of DB-lipidoids
DB-lipidoids were synthesized using a one-pot, two-step method. First, amine (0.1 mmol, 1 equiv.) and epoxide (0.24 mmol, 2.4 equiv.) were combined in a glass vial and heated at 80 °C for 48 h. Next, the reactant was dissolved in 2 mL anhydrous DCM at RT, followed by the addition of acyl chloride (0.24 mmol, 2.4 equiv.) and TEA (0.3 mmol, 3 equiv.). 12 h later, DCM and TEA were removed under vacuum and crude DB-lipidoids were dissolved in EtOH for initial screening. Notably, if an amine was in the salt form, excessive TEA was added to neutralize it in the first step. All crude DB-lipidoids were confirmed by mass spectrometry (Table S1).

## Purification and characterization of 11-10-8
To purify the top-performing DB-lipidoid 11-10-8, its crude product was separated using a CombiFlash NextGen 300+ chromatography system with gradient elution from $CH_2Cl_2$ to 75:22:3 $CH_2Cl_2$/MeOH/NH$_4$OH (aq), and the desired fractions were collected as light brown oil (yield 82%). 11-10-8 was characterized by mass spectrometry (MS) and nuclear magnetic resonance spectroscopy (NMR). MS-ESI: calculated for $C_{42}H_{82}N_2O_4$: 678.63, found [M + H]$^+$ = 679.62; $^1H$ NMR (400 MHz, MeOD) δ: 4.95 (tt, J = 8.2, 4.3 Hz, 2H), 3.19 – 3.08 (m, 2H), 2.84 (t, J = 6.9 Hz, 2H), 2.72 (dd, J = 14.0, 7.7 Hz, 2H), 2.62 (ddd, J = 14.0, 10.3, 4.0 Hz, 2H), 2.45 – 2.28 (m, 4H), 2.14 – 2.04 (m, 4H), 1.72 – 1.48 (m, 8H), 1.42 – 1.26 (m, 44H), 0.93 (h, J = 3.1 Hz, 12H).

## LNP formulation and optimization
For initial in vitro and in vivo screening, DB-LNPs were prepared by pipette mixing of the ethanolic phase containing DB-lipidoid, DOPE, cholesterol and DMG-PEG with an aqueous phase (10 mM citrate buffer, pH 3) containing mRNA at a volume ratio of 1:3 and then diluted in culture medium or 1× PBS for cell or animal treatment, respectively. The weight ratio of DB-lipidoid, DOPE, cholesterol, DMG-PEG and mRNA was fixed at 16:10:10:3:1.6. The mRNA encapsulation efficiency was typically 40-80%.

To formulate 11-10-8 DB-LNP by microfluidic mixing, the ethanolic phase containing lipids was mixed with the aqueous phase containing mRNA at a flow rate ratio of 1:3 and at a 11-10-8/mRNA weight ratio of 10:1 in a microfluidic chip device[46]. To optimize the formulation of 11-10-8 DB-LNP, various LNPs formulated by microfluidic mixing were

tested in vivo, and the optimal one with a molar ratio of 11-10-8/DOPE/Chol/DMG-PEG at 40:10:48.8:1.5 was chosen for subsequent studies. The benchmark MC3 LNP (or SM-102 LNP) was formulated with MC3 (or SM-102), DSPC, cholesterol and DMG-PEG at a molar ratio of 50:10:38.5:1.5 using microfluidic mixing at an MC3/mRNA weight ratio of 10:1. LNPs were dialyzed against 1× PBS in a 20 kDa MWCO cassette for 2 h, filtered through a 0.22 μM filter and stored at 4 °C.

## Characterization
$^1H$ NMR were recorded using a Bruker 400 MHz NMR spectrometer. LC-MS was performed on a Waters Acquity LCMS system equipped with UV-Vis and MS detectors. The hydrodynamic size, polydispersity index (PDI) and zeta potential of LNPs were measured using a Malvern Zetasizer Nano ZS90. The morphology of LNPs was characterized by a cryo-electron microscope (Titan Krios, Thermo Fisher). The mRNA encapsulation efficiency and the p$K_a$ of LNP were determined using a modified Quant-iT RiboGreen RNA assay (Invitrogen, Carlsbad, California, USA) and a 6-(p-toluidinyl)naphthalene-2-sulfonic acid (TNS) assay[47,48], respectively.

## Cell culture and animal studies
Human hepatocellular carcinoma HepG2 cell line was purchased from American Type Culture Collection (#HB-8065, ATCC, Manassas, Virginia, USA) and maintained in Dulbecco's Modified Eagle Medium (DMEM) supplemented with 10% fetal bovine serum (FBS), 100 U/mL penicillin and 100 μg/mL streptomycin. Cells were cultured at 37 °C in a humidified incubator of 5% $CO_2$, and routinely tested for mycoplasma contamination.

All animal protocols were approved by the Institutional Animal Care and Use Committee (IACUC) of the University of Pennsylvania (Protocol No. 806540), and animal procedures were performed in accordance with the Guidelines for Care and Use of Laboratory Animals at the University of Pennsylvania. C57BL/6 female mice (6-8 weeks, 18-20 g) and C57BL/6 male mice (6-8 weeks, 22-24 g) were purchased from Jackson Laboratory.

## High-throughput in vitro and in vivo screening of DB-LNPs
HepG2 cells were seeded in 96-well plates at a density of 5000 per well overnight and mLuc-loaded DB-LNPs at a dose of 15 ng mRNA/well (0.24 nM) were used to treat cells for 24 h. Luciferase expression was evaluated by Luciferase Reporter 1000 Assay System (#E4550, Promega, Madison, Wisconsin) and cell viability was measured using a CellTiter-Glo Luminescent Cell Viability Assay (#G7572, Promega) according to manufacturer's protocols. For in vivo screening, female mice were i.v. injected with mLuc-loaded DB-LNPs at an mRNA dose of 0.1 mg/kg. 4 h later, mice were intraperitoneally (i.p.) injected with D-luciferin potassium salt (150 mg/kg), and bioluminescence imaging was performed using an in vivo imaging system (PerkinElmer).

## Molecular dynamics simulations
Lipid dynamics simulations of 11-10-8 and MC3 were optimized at CHARMm force field[49]. The lipid simulations were run up to 4000 steps to achieve the energy-minimized structures. All bonds containing intermolecular interactions were constrained using the Smart Minimizer algorithm (i.e., 1000 steps of Steepest Descent with Root-Mean-Squared (RMS) gradient of 3 and Conjugate Gradient minimization). The overall RMS gradient tolerance was set to 0.01. Momany-Rone method was assigned for partial charge estimation using Discovery Studio 2018 (Accelrys)[49]. The dimensionless packing parameter *P* of a lipid molecule was calculated as $P = V/(AL)$ based on its Van der Waals molecule volume (*V*), cross section area of polar head (*A*) and average tail length (*L*)[34]. *V*, *A* and *L* were derived from the optimized conformation of the lipid as well as the estimated atomic Van der Waals radius[50].

## Hemolysis assay

Mouse red blood cells (RBCs) were isolated and washed three times with 1× PBS by centrifugation at 700 g for 5 min. Next, RBCs were diluted to a 4% vol/vol RBC suspension either in neutral (pH 7.4) or acidic PBS (pH 6.0), and incubated with LNPs at a final mRNA concentration of 3 µg/mL at 37 °C for 1 h. Finally, the RBC suspension was centrifuged at 700 g for 5 min and 100 µL supernatant was transferred into a 96-well plate. The absorption at 540 nm was determined with a plate reader. Positive and negative controls were carried out with 0.1% Triton-X and 1× PBS, respectively.

## Cellular internalization inhibition

HepG2 cells were seeded in a 96-well plate at a density of 5,000 per well overnight. Cells were pre-treated with 5 mM amiloride, 20 µM chlorpromazine, 0.2 mM genistein or 5 mM Mβ-CD for 30 min. Then, cells were treated with mLuc-loaded LNPs (15 ng/well) for 24 h. Luciferase expression was determined as described above.

## Systemic mRNA delivery

For GFP mRNA delivery, female mice were i.v. injected with GFP mRNA-loaded LNPs at an mRNA dose of 0.25 mg/kg. 4 h later, mice were euthanized and major organs (heart, liver, spleen, lung, and kidneys) were collected for ex vivo fluorescence imaging. Livers were collected for cryosectioning and samples were incubated with vascular endothelial cadherin (VE-Cad) antibody (1:200, AF1002, R&D, Minneapolis, Minnesota, USA) and F4/80 antibody (1:200, #30325, CST, Danvers, Massachusetts, USA) overnight at 4 °C. After being washed three times, samples were incubated with Alexa Fluor™ 647 conjugated donkey anti-goat IgG (H + L) cross-adsorbed secondary antibody (1:500, A-21447, Invitrogen) and Alexa Fluor™ 568 conjugated donkey anti-rabbit IgG (H + L) cross-adsorbed secondary antibody (1:500, A10042, Invitrogen) for 1 h at RT. Nuclei were stained with DAPI (10 µg/mL) before images were taken using a confocal laser scanning microscope (LSM 710, Zeiss).

For Cas9 mRNA/TTR sgRNA co-delivery, female mice were i.v. injected with Cas9 mRNA/sgRNA (4:1, wt:wt)-loaded LNPs at a total RNA dose of 1 mg/kg. Serum was collected at on day 0, 3, 7, 14, 28 and analyzed by ELISA (Aviva Systems Biology, #OKIA00111). Some of mice were euthanized on day 7, and livers were collected to determine the on-target indel frequency by next-generation sequencing (NGS). For TTR on-target DNA sequencing, DNA was extracted from the liver using the Qiagen Puregene Tissue Kit (#158063) and quantified using a Nanodrop 2000. PCR amplification of the TTR target site was carried out using Q5 High-Fidelity DNA Polymerase (New England Biolabs, #M0491) and the following primer sequences: mTTR-exon2-F, 5'-CGGTTTACTCTGACCCATTTC-3' and mTTR-exon2-R, 5'-GGGCTTTCTACAAGCTTACC-3'. Deep sequencing of the TTR amplicons and determination of the on-target indel frequency was performed essentially as described except that 150 bp pair end reads were produced[51].

## Safety evaluation

Serum was collected at 24 h post-treatment of Cas9 mRNA/TTR sgRNA-loaded LNPs (1 mg/kg). Liver function was evaluated by measuring serum AST and ALT activities according to the manufacturer's instruction. 13 mouse cytokines were examined using a LEGENDplex™ multi-analyte flow assay kit (#740621, BioLegend, San Diego, California, USA) according to the manufacturer's instruction.

## FGF21 mRNA therapy in obese mice

Male mice were fed a HFD (#MP290194410, MP Biomedicals, Santa Ana, California, USA) for three weeks to induce obesity (body weight ~30 g) and fatty liver[41]. These obese mice were i.v. injected with FGF21 mRNA-loaded LNPs at an mRNA dose of 0.25 mg/kg. Serum was collected at 0, 3, 6, 12, 24 and 48 h post-injection and analyzed by ELISA (R&D, #DF2100).

For the therapeutic study, obese mice were i.v. injected with various LNP formulations at an mRNA dose of 0.25 mg/kg every other day for three doses. Body weights were recorded every two days since the start of the first treatment. Male mice fed with a normal chow diet (NCD) was used as a control group. At the end of the experiment, mice were anesthetized, and livers were excised, rinsed, and weighed. Livers were fixed in 10% neutral formalin overnight, embedded in paraffin, cut into 5 µm sections, and stained with hematoxylin and eosin (H&E) for histological examination of steatosis. The semi-quantitative analysis for liver steatosis was performed according to an adapted scoring protocol[52].

## Systemic siRNA delivery

Three TTR siRNAs were pooled at a 1:1:1 molar ratio. Female mice were i.v. injected with TTR siRNA-loaded LNPs at a total siRNA dose of 1 mg/kg. Serum was collected at 0, 3, 7, 14, 21, and 28 days post-injection and analyzed by ELISA. To determine the ED50, mice were i.v. injected with TTR siRNA-loaded LNPs at a total siRNA dose of 0.01, 0.02, 0.05, 0.1 or 1 mg/kg. On day 3, serum was collected for ELISA analysis.

## Intramuscular mRNA delivery

Mice were i.m. injected with mLuc-loaded LNPs at an mRNA dose of 0.1 mg/kg. 4 h later, mice were i.p. injected with D-luciferin potassium salt (150 mg/kg), and bioluminescence imaging was performed using an in vivo imaging system.

## Statistical analysis

Data are presented as mean ± SD. Student's t-test or one-way analysis of variance (ANOVA) followed by Tukey test was applied for comparison between two groups or among multiple groups using Graphpad Prism 8.0, respectively. $p < 0.05$ was considered to be statistically significant.

## Reporting summary

Further information on research design is available in the Nature Portfolio Reporting Summary linked to this article.

## Data availability

DNA sequencing files can be accessed at the National Center for Biotechnology Information Sequence Read Archive (NCBI SRA) with accession code "PRJNA1064156". All other data supporting the findings of this study are available within the article and its supplementary files. Any additional requests for information can be directed to, and will be fulfilled by, the corresponding author. Source data are provided with this paper.

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

## Acknowledgements

M.J.M. acknowledges support from a US National Institutes of Health (NIH) Director's New Innovator Award (DP2 TR002776), a Burroughs Wellcome Fund Career Award at the Scientific Interface (CASI), a US National Science Foundation (NSF) CAREER Award (CBET-2145491), and a grant from the American Cancer Society (RSG-22-122-01-ET). M.J.M. and J.M.W. acknowledge support from a sponsored research agreement with iECURE. The authors acknowledge Dr. Stefan Steimle from the Beckman Center for Cryo Electron Microscopy at UPenn for the help in characterizing the morphology of LNPs. The authors acknowledge UPenn Gene Therapy Program NAT Core for sequencing service and thank Kelly Martins for processing the NGS data. The authors acknowledge Dr. Yi Zhong from the State Key Laboratory of Natural and Biomimetic Drugs, Peking University for providing access to computing resources in molecular dynamic simulation studies.

## Author contributions

Conceptualization: X.H., Y.X., M.J.M.; Methodology: X.H., J.X., M.G.A.; Investigation: X.H., J.X., M.G.A., G.Z.; Visualization: X.H., J.X., Y.X.; Funding acquisition: M.J.M.; Supervision: M.J.M.; Writing—original draft: X.H., J.X., Y.X.; Writing—review & editing: J.X., Y.X., M.G.A., L.X., N.G., R.E.M., R.P., G.Z., A.E.V., C.C.W., J.M.W., D.W., M.J.M.

## Competing interests

X.H. and M.J.M. are inventors on a patent filed by the Trustees of the University of Pennsylvania (U.S. Provisional Patent Application No. 63/581,832, filed September 11, 2023) describing degradable branched lipid nanoparticle technology in this manuscript. JMW is a paid advisor to and holds equity in iECURE, Scout Bio, Passage Bio, and the Center for Breakthrough Medicines (CBM). He also holds equity in the former G2 Bio asset companies. He has sponsored research agreements with Amicus Therapeutics, CBM, Elaaj Bio, FA212, Foundation for Angelman Syndrome Therapeutics, former G2 Bio asset companies, iECURE, Passage Bio, and Scout Bio, which are licensees of Penn technology. J.M.W. is an inventor on patents that have been licensed to various biopharmaceutical companies and for which he may receive payments. All other authors declare no competing interests.
