## [Peer review file · Nature Communications]

REVIEWER COMMENTS

Reviewer #1 (Remarks to the Author):

In this manuscript, Han, Xu et al. report the generation of SRB-LNPs and claim that 11-10-8 LNP have higher efficacy when compared to MC3. To support their claims, authors delivered FGF21 mRNA and siRNA against TTR and conducted preliminary studies. In general, manuscript is well-written and present advancement to the field of delivery of RNA therapeutics for liver disorders. Overall, the presented data to some extent support authors' claims, however require further validation in other mouse models and delivery of other types of RNA. I have following comments to improve the manuscript further.

Major comments:

1. Authors delivered FGF21 mRNA using 11-10-8 LNP or MC3 LNP in wild-type mice. As authors have mentioned in the manuscript, FGF21 is a promising target for NASH, and in fact it is under clinical trials as well. Therefore, the experiments presented in figure 5 should be performed in mice fed with high-fat diet to at least mimic a few features of NAFLD / NASH. This would ascertain the relevance of 11-10-8 LNP towards clinical application.
2. The last line of the results section "These results suggest that our SRB-LNPs could be a universal platform for robust delivery of various RNA constructs". I consider this is one of the strengths of this study. In fact, having a universal LNP system for different types of RNA would be extremely helpful for variety of applications. However, to prove their such statement authors need to demonstrate the delivery of another RNA (such as microRNA) molecule using their 11-10-8 LNP in addition to already presented large mRNA (FGF21) and siRNA (TTR). The data showing the miRNA delivery using their 11-10-8 LNP would raise the impact of authors' current study.
3. The data on intrahepatic distribution of 11-10-8 GFP-LNP is missing. Specifically, GFP fluorescence should be presented with cell type-specific staining such as for cholangiocytes, hepatic stellate cells, Kupffer cells, macrophages and liver sinusoidal endothelial cells.

Minor comments

1. (Line 314, 315, 316) Other internalization pathways (e.g., macropinocytosis, clathrin-mediated endocytosis and caveolae-mediated endocytosis) were also involved but contributed less to overall LNP internalization. No data has been cited in the paper for this. Is there any data for this claim?
2. Which silica-based columns were used for the removal of double stranded RNA?
3. Did the authors determine capping efficiency? If so, how was it determined? What is the percentage of capped mRNA?
4. For GFP and FG21 mRNA delivery, mice were i.v. injected with GFP mRNA-loaded LNPs at an mRNA dose of 0.25 mg/kg, how did the authors come up with this dose? Did the authors try other doses?

Reviewer #2 (Remarks to the Author):

General comments:

This is an interesting and systematic study reporting the effect of structure of branched lipidoids on the ability to delivery mRNA, Cas9/sgRNA and siRNA. However, the manuscript has a number of flaws that require the author's attention.

What determines the delivery efficiency is not only the chemical structure of the ionizable lipid, but also the colloidal carrier. From the drawn structure of the lipidoids, it makes sense to use the space shuttle analogy. However, the authors do not actually show that the intracellular degradation of the branched tails cause dissociation and release of the cargo, which makes the space shuttle analogy rather speculative and more a selling point of the paper than solid hypothesis-based research. The endosomal escape and mRNA release may rather be the result of a structural change in the nanoparticles caused by the charge reversal and the DOPE component of the nanoparticles. I suggest that the authors take the space shuttle analogy out, unless they have more solid experimental proof for their space shuttle hypothesis.

The authors are using the terms ionizable lipids and lipidoids alternately without defining them. It is an interesting point to discuss the nomenclature of lipid nanoparticles and the term lipidoid. It seems that the term "lipidoid" is most appropriate as these materials are lipid-like, and not truly lipids as defined previously (A combinatorial library of lipid-like materials for delivery of RNAi therapeutics | Nature Biotechnology) and also in this manuscript. However, they can also be termed as synthetic amino lipids (a term that Moderna also uses: A Novel Amino Lipid Series for mRNA Delivery: Improved Endosomal Escape and Sustained Pharmacology and Safety in Non-human Primates: Molecular Therapy (cell.com)).

The authors have chosen to use MC3/the Onpattro formulation for reference purposes, which was developed and optimized specifically for delivery of small interfering RNA to the liver. Although MC3 is used in general by many research groups as the 'benchmark' ionizable lipid OR lipidoid (in this context), it was not to be used as a benchmark for mRNA delivery as stated in a review by the pioneers in the field (Lipid Nanoparticle Systems for Enabling Gene Therapies: Molecular Therapy (cell.com)). For example, plasmid DNA delivery is greatly enhanced by DLin-K2-DMA instead of MC3-DMA (In vivo delivery of plasmid DNA by lipid nanoparticles: the influence of ionizable cationic lipids on organ-selective gene expression - Biomaterials Science (RSC Publishing)). Thus, in this case, if the delivery of plasmid DNA is to be considered, the benchmark would be KC2-DMA. Hence, MC3 is not the right benchmark lipid to use for mRNA delivery. Therefore, the authors should compare with the gold standard ionizable cationic lipids used in the COVID-19 vaccines, i.e. SM102 and/or ALC-0315, and compare with these lipids using the i.m. route of administration.

For library 1, the authors should report the physicochemical properties of the LNPs. It is unlikely that all structures are equally well suited for forming nanoparticles using the same LNP composition for all formulations, and it is relevant to report also how the lipidoid structure influences the colloidal stability of the LNPs, also in the presence of serum proteins. The failure of some of the formulation could be due to suboptimal compositions and not (only) the structure of the lipidoid.

For all concentrations, please convert ng to nmol.

For many experiments, $n = 2$. It is not possible to report a standard deviation, if $n = 2$. Please report mean values \pm standard deviations of at least triplicates.

Specific comments:

Title:

The title is misleading, because it is not the lipid nanoparticles that are branched, but the ionizable lipids used to form the nanoparticles. In addition, potency is an intrinsic drug property, not a property of the transporter. The nanoparticle transporters should rather be described as efficient, because they are mediating a process. Please correct accordingly.

Abstract:

Line 34: Please rephrase "synthetic challenges"

Line 38 and throughout the manuscripts: Numbers 1-9 are written as words (also page 3, line 107)

Line 46: Replace "potent" with "efficient" (see comment to the title)

Introduction:

Page 2, line 49: To date the mRNA technology is only used in two approved products, i.e. the COVID-19 mRNA vaccines. Hence, it has only enable breakthroughs in the prevention of COVID-19. Please modify the text to reflect this.

Page 2, line 51: Again, the authors are overselling the mRNA technology by stating "tremendous clinical success". Only two products are approved. Please modify the text accordingly.

Page 2, line 57: The authors are using the terms ionizable lipids and lipidoids alternately without defining them. See comment above about the use of the terms ionizable lipids and lipidoids. For this manuscript, is there a distinction between ionizable lipids (or lipidoids) or should it be ionizable lipids (also termed as lipidoids synonymously)? Please define what lipidoids are, and make clear what the difference is between lipidoid and ionizable lipids.

Page 3, line 77: The authors have summed up the references for use of the branched tail

intermediates coupled to functional groups, but it is also vital to cite the literature for studies related to the headgroups. Especially since the crucial amine 11 from this study was the same as amine 76 from one of the earliest studies (A combinatorial library of lipid-like materials for delivery of RNAi therapeutics | Nature Biotechnology (Figure 1). In addition, many of the amines in this manuscript correspond to the amines used in the above study. Amine 1 from the manuscript is amine 80, amine 6 is amine 10, and so on.

Page 3, lines 100-101: MC3-based LNPs are clinically approved for siRNA, not mRNA. Please modify the text accordingly.

Results:

Page 4, line 117: Part of the structure is not degradable. Please modify the text to reflect this.

Page 4, line 128: The authors argue that the crude lipidoids can be used directly for LNP formulation without additional purification, but the argumentation is not clear. If the two reactions are 80% efficient, the combined efficiency is $0.8 \times 0.8 = 64\%$. How can the authors be sure that impurities are not affecting the LNP formulation?

Page 6, line 152: MC3 was developed for siRNA delivery. Hence, "gene delivery" should be corrected to "siRNA delivery".

Page 6, lines 152-155: The authors aim to minimize the size of the non-degradable parts "for accelerated physiological clearance". Is that achieved?

Page 6, line 161: "25 SRB-LNPs" should be "25 SRB-LNP formulations". Please provide the molar ratios of the LNP components.

Page 6, lines 183-4: The sentence "Since the total C number in the tail region is closely related to the hydrophobicity, it is reasonable to see the optimal one to be 18". The authors should explain why it is reasonable, and this should be moved to the discussion.

Page 7, line 198: "mRNA delivery potency" should be corrected to "mRNA delivery efficiency"

Page 7, line 205: "Ingredient" should be "agent" or "excipient".

Page 7, lines 209-211: The authors should show that intracellular branch tail degradation causes dissociation and mRNA release, and not just speculate. This is the basis for the space shuttle analogy.

Page 7, line 217: Do not start a sentence with a number.

Page 9, line 251: Is the 5-fold difference claimed by the authors statistically significant?

Page 10, line 295: Again, is the 7-fold difference statistically significant?

Page 10, line 298: There is a typo in physiochemical, which should be physicochemical.

Page 10, line 304: Avoid non-scientific and relative words like "good".

Page 12, lines 336-338: Although the route of administration was kept constant (i.v.) by comparing to MC3, it would have been a fair comparison, if the 11-10-8 LNP was administered subcutaneously or intramuscularly and compared head-to-head to Lipid 5 or Lipid H (SM-102), which is relevant for mRNA delivery.

Page 12, line 345-347: Please adjust the number of significant figures. Commas should be corrected to full stops.

Page 12, line 365: The authors claim equal potency. This should be confirmed statistically

Page 13, last paragraph: This section should be conclusions, not just a summary.

Materials and methods

Please include city and country of the headquarters of all vendors.

Page 14, line 454: Please explain "siRNA pool". Is it a 1:1:1 molar ratio of the three different siRNAs? The sequences should be provided.

Page 14, line 489: Is this ratio weight ratio or molar ratio? Please provide the lipid molar ratio (in percent) to enable direct comparison with the MC3 formulation + lipid:mRNA weight ratio.

Page 14, line 495. Is this the molar ratio? Please state that.

Figures:

Figure 1: This figure is rather speculative and indeed misleading, because the authors are not showing in the manuscript that nanoparticle disassembly is caused by the cleavage of the ester bonds in the ionizable lipids. The endosomal escape and mRNA release may rather be the result of a structural change in the nanoparticles caused by the charge reversal and the DOPE component of the nanoparticles. I suggest that the authors take the figure out, unless they have more solid experimental proof for their space shuttle hypothesis.

Figure 2: Please include mean values for triplicates in c) and d), and state in the legend that mean values are reported. For c), please state the cell type that is used.

Figure 3: What is shown in b) and c)? Mean values? Please state that in the legend. It is not possible to have standard deviation when $n = 2$. The in vivo luminescence background in this graph is 107, while that in Figures 4a and 4b is 105. This is a huge difference in terms of the values, because the total flux for the MC3-based formulation is practically the same. It would be great to know the author's input on this.

Figure 4: For the in vivo experiments, $n = 2$, which is too little for meaningful data, considering the biological variation. Please include more replicates for proper statistics. There is a typo in zet potential. In the legend, there is a typo in physiochemical, which should be physicochemical. In the same line "parameters" should be "properties". "Lipid rafter-mediated" should be "lipid raft-mediated".

Figure 5: Although $n = 3$ is considered as the minimum requirement for applying statistics, based on the deviations for MC3 in Figure 5d, it is appropriate to include more number of samples for analysis. For the graphs in c) and e), data points should not be connected with full lines. Full lines are reserved for mathematical fits or continuous data sets. More data points should be included.

Supplementary data:

Figure S1: Please include the structure of MC3 for comparison. What is the biophysical rationale for the design of these structures?

Figure S2: Legend of x-axis: What does "experiment no" refer to? Which structure? It is not possible to have standard deviation when $n = 2$. The dose should be given in nmol, not ng. The same comments go for Figure S7.

Figure S11: The polynomial fit is wrong. Data should be fitted to a sigmoidal curve.

Figure S16: Three data points are not enough for proper determination of ED50 values. Please include more data points.

Table S2: Are the same formulations tested for siRNA, Cas9/sgRNA and mRNA? What are the N/P ratios (or weight ratios of nucleic acid cargo)?

Table S4: The plasma concentration profile has only three data point, of which one point is very low. Therefore, there are not enough data points on this curve to justify determination of the AUC values given in Table S4.

Reviewer #3 (Remarks to the Author):

I co-reviewed this manuscript with one of the reviewers who provided the listed reports as part of the Nature Communications initiative to facilitate training in peer review and appropriate recognition for co-reviewers.

Reviewer #4 (Remarks to the Author):

Focus of this manuscript is majorly chemistry and optimization. Seems the content is best focus for chemistry journal like ACIE, Chem Sci, JACS, JCR.

Authors may explain better how the molecule is like space shuttle. Chemical design are popular amine-epoxide ring opened lipidoids with hydroxyl position modified to add more hydrophobic tails. The carton look like LNP with rocket ship conjugated to the surface like targeting ligand, but actual structure is regular LNP structure. This is little confusing.

Also other papers show role of branched tails in mRNA delivery already published. Hashiba et al. "Branching Ionizable Lipids Can Enhance the Stability, Fusogenicity, and Functional Delivery of mRNA" *Small Science*, Volume 3, 2200071, 2023. Hajj et al. "Branched-Tail Lipid Nanoparticles Potently Deliver mRNA In Vivo due to Enhanced Ionization at Endosomal pH" *Small*, 15, 1805097, 2019. Hajj et al. "A Potent Branched-Tail Lipid Nanoparticle Enables Multiplexed mRNA Delivery and Gene Editing In Vivo" *Nano Letters* 20, 5167, 2020. Sabnis et al. "A Novel Amino Lipid Series for mRNA Delivery: Improved Endosomal Escape and Sustained Pharmacology and Safety in Non-human Primates" *Molecular Therapy*, 26, 1509, 2018.

REVIEWERS' COMMENTS AND AUTHORS' ANSWERS

Note: Our responses (standard typeface) to reviewers' comments (bold); the yellow highlighted words and sentences have been added to the main text.

Reviewer #1 (Remarks to the Author):

In this manuscript, Han, Xu et al. report the generation of SRB-LNPs and claim that 11-10-8 LNP have higher efficacy when compared to MC3. To support their claims, authors delivered FGF21 mRNA and siRNA against TTR and conducted preliminary studies. In general, manuscript is well-written and present advancement to the field of delivery of RNA therapeutics for liver disorders. Overall, the presented data to some extent support authors' claims, however require further validation in other mouse models and delivery of other types of RNA. I have following comments to improve the manuscript further.

Response: We thank the reviewer for this important feedback. We also thank the reviewer for their time and effort in helping us improve the manuscript. We were excited to hear that the reviewer felt that our manuscript was well-written and presented advancement to the field of delivery of RNA therapeutics for liver disorders. We have now performed more experiments to support our claims and address the reviewer's concerns. We hope that the reviewer enjoys our revised manuscript.

Major comments:

1. Authors delivered FGF21 mRNA using 11-10-8 LNP or MC3 LNP in wild-type mice. As authors have mentioned in the manuscript, FGF21 is a promising target for NASH, and in fact it is under clinical trials as well. Therefore, the experiments presented in figure 5 should be performed in mice fed with high-fat diet to at least mimic a few features of NAFLD / NASH. This would ascertain the relevance of 11-10-8 LNP towards clinical application.

Response 1: We thank the reviewer for the great suggestion. We have now evaluated 11-10-8 LNP for the delivery of FGF21 mRNA in high fat diet (HFD)-induced obese mice and assessed the therapeutic potential of 11-10-8 LNP-based FGF21 mRNA therapy for the treatment of obesity and fatty liver. The results showed that FGF21 mRNA-loaded 11-10-8 LNP enabled higher expression of FGF21 and demonstrated stronger weight-reducing and lipid-lowering effects than MC3 LNP in obese mice. We have now updated these results and discussion in the section **2.7 Hepatic delivery of mRNA-based therapeutics using DB-LNPs** (Page 11, line 365) as follows: "Hepatic delivery of mRNA-based therapeutics holds great promise for protein supplementation therapies. FGF21 is a pleiotropic metabolic hormone primarily secreted by the liver, which is a promising therapeutic agent for obesity, type 2 diabetes and non-alcoholic steatohepatitis^{39, 40}. We next evaluated 11-10-8 LNP for the

delivery of human FGF21-encoded mRNA in high fat diet (HFD)-induced obese mice. Male mice were fed a HFD for three weeks to induce obesity (body weight ~30g) and fatty liver⁴¹, and then i.v. injected with FGF21 mRNA-loaded LNPs. The expression of FGF21 peaked (58770 ± 1087 pg/mL) at 12 h post-administration of FGF21 mRNA-loaded 11-10-8 LNP, which was five-fold higher than that in FGF21 mRNA-loaded MC3 LNP-treated mice (11764 ± 1064 pg/mL, **Figure 6a**). Moreover, the area under curve (AUC) – a pharmacokinetic metric of therapeutic exposure – of FGF21 was 4.6-fold greater in 11-10-8 LNP-treated mice than that in MC3 LNP-treated mice (**Table S5**).

To further demonstrate the therapeutic potential of this FGF21 mRNA therapy, we examined its weight-reducing and lipid-lowering effects in obese mice³⁹. Obese mice were treated with various LNP formulations every other day for three doses (**Figure 6b**). While obese mice treated with PBS or mLuc-loaded LNPs gradually increased weight, obese mice treated with FGF21 mRNA-loaded 11-10-8 LNP or FGF21 mRNA-loaded MC3 LNP lost weight or maintained weight, respectively (**Figure 6c**). At the end of this experiment, body weight as well as liver weight of obese mice treated with FGF21 mRNA-loaded 11-10-8 LNP was significantly reduced compared to that of obese mice treated with other formulations (**Figure 6d,e**). Hematoxylin and eosin (H&E) staining of livers showed less vacuoles and reduced liver steatosis in the obese mice treated with FGF21 mRNA-loaded 11-10-8 LNP compared to those obese mice treated with other formulations (**Figure 6f,g**). It is worth mentioning that although MC3 LNP-based FGF21 mRNA therapy exhibited weight-reducing and lipid-lowering effects to some extent in obese mice, they were less obvious than 11-10-8 LNP-based therapy, presumably due to the less FGF21 expression (**Figure 6a**). Together, these results suggest that 11-10-8 LNP-based FGF21 mRNA therapy could be a promising approach for treating obesity and fatty liver.

Figure 6. DB-LNP-mediated hepatic delivery of FGF21 mRNA. (a) LNP-mediated FGF21 mRNA delivery ($n = 4$). Male mice were fed a HFD for three weeks to induce obesity (body weight ~ 30 g) and fatty liver. These obese mice were i.v. injected with FGF21 mRNA-loaded LNPs at an mRNA dose of 0.25 mg/kg. Serum was collected at the indicated time points for ELISA analysis of FGF21. (b) A scheme of FGF21 mRNA therapy in HFD-induced obese mice. Obese mice were i.v. injected with various LNP formulations at an mRNA dose of 0.25 mg/kg every other day for three doses. Male mice fed a normal chow diet (NCD) was used a control group. (c) Body weight growth curve ($n = 4$). (d) Body weight on Day 6 ($n = 4$). (e) Liver weight ($n = 4$). (f) Representative images of liver histological examinations (H&E staining). Scale bars: 100 μ m. (g) Liver steatosis score ($n = 12$). A total of 12 images for each group (three random fields for each liver section, four mice per group) were analyzed semi-quantitatively for liver steatosis. Data are presented as mean \pm SD.

2. The last line of the results section “These results suggest that our SRB-LNPs could be a universal platform for robust delivery of various RNA constructs”. I consider this is one of the strengths of this study. In fact, having a universal LNP system for different types of RNA would be extremely helpful for variety of applications. However, to prove their such statement authors need to demonstrate the delivery of another RNA (such as microRNA) molecule using their 11-10-8 LNP in addition to already presented large mRNA (FGF21) and siRNA (TTR). The data showing the miRNA delivery using their 11-10-8 LNP would raise the impact of authors’ current study.

Response 2: We thank the reviewer for these comments, and apologize for making

this overstatement. We agree with the reviewer that robust delivery of various RNA molecules (e.g., miRNA and ASO) must be demonstrated in order to make such a statement. Our study is focused on mRNA delivery, and we have successfully shown that our 11-10-8 LNP outperformed MC3 LNP in the delivery of various mRNAs including luciferase mRNA, GFP mRNA, FGF21 mRNA and Cas9 mRNA/sgRNA. We tested siRNA in our study because the benchmark MC3 LNP is FDA-approved for *TTR* silencing, and it was interesting to assess whether our 11-10-8 LNP outperforms this industry standard LNP in terms of siRNA delivery.

Therefore, we have now reduced our claims as we believe that it is beyond the scope of our study to test miRNAs as suggested by the reviewer. Moreover, as far as we know, there are no miRNAs that can specifically target *TTR* mRNA to inhibit its translation. We have now rephrased this claim (Page 13, line 413) to avoid the overstatement as follows: “These results suggest that our DB-LNPs could also be a robust platform for the delivery of siRNA.”

3. The data on intrahepatic distribution of 11-10-8 GFP-LNP is missing. Specifically, GFP fluorescence should be presented with cell type-specific staining such as for cholangiocytes, hepatic stellate cells, Kupffer cells, macrophages and liver sinusoidal endothelial cells.

Response 3: We thank the reviewer for this important suggestion. We have now performed immunofluorescence staining of Kupffer cells and liver sinusoidal endothelial cells, two main non-parenchymal cells in the liver. These data are presented in **Fig. S17** and the corresponding discussion has now been updated in the manuscript (Page 10, line 339) as follows: “Immunofluorescence staining results further confirmed that 11-10-8 LNP mediated greater GFP expression than MC3 LNP in major liver cells, including hepatocytes, Kupffer cells and liver sinusoidal endothelial cells (**Fig. S17**).”

Figure. S17 GFP expression in the liver at 4 h post-treatment of GFP mRNA-loaded LNPs. Mice were i.v. injected with GFP mRNA-loaded LNPs at an mRNA dose of 0.25mg/kg. Livers were collected and cryo-sectioned for immunofluorescence staining at 4 h post-treatment. Kupffer cells (F4/80⁺) and liver sinusoidal endothelial cells (VE-Cad⁺) were stained, respectively. Scale bars: 100 μ m.

Minor comments

1. (Line 314, 315, 316) Other internalization pathways (e.g., macropinocytosis, clathrin-mediated endocytosis and caveolae-mediated endocytosis) were also involved but contributed less to overall LNP internalization. No data has been cited in the paper for this. Is there any data for this claim?

Response 1: We thank the reviewer for raising this question, and apologize for not properly citing the data. We have now cited the data and rephrased this claim (Page 10, line 319) as follows: “Other internalization pathways (e.g., macropinocytosis, clathrin-mediated endocytosis and caveolae-mediated endocytosis) were also involved but contributed less to overall LNP internalization, since their inhibitors only slightly suppressed mLuc delivery (Figure 4e).”

2. Which silica-based columns were used for the removal of double stranded RNA?

Response 2: We thank the reviewer for this question. We used cellulose-based columns to remove double-stranded RNAs based on a method published by Katalin Karikó *et al* (*Molecular Therapy-Nucleic Acids*, 2019, 15: 26-35), where this process was described in detail. We have now cited this paper and updated this information in the section **4.2. mRNA Synthesis**.

3. Did the authors determine capping efficiency? If so, how was it determined? What is the percentage of capped mRNA?

Response 3: We thank the reviewer for these questions. We used TriLink’s proprietary co-transcriptional capping reagent (#N-7113, TriLink) for *in vitro* transcription of 5’ capped mRNA (CleanCap™ technology). This CleanCap™ technology has shown to provide up to 98% capping efficiency (<https://www.trilinkbiotech.com/cleancap-reagent-ag.html>). We consistently achieved above 95% capping efficiency for various mRNAs using this technology in our practice. The capping efficiency was determined by HPLC.

4. For GFP and FG21 mRNA delivery, mice were i.v. injected with GFP mRNA-loaded LNPs at an mRNA dose of 0.25 mg/kg, how did the authors come up with this dose? Did the authors try other doses?

Response 4: We thank the reviewer for these questions. Apart from 0.25 mg/kg, we also tried a lower dose (0.05 mg/kg) and a higher dose (1 mg/kg) in our pilot study, and found that 11-10-8 LNP consistently outperformed MC3 LNP in GFP mRNA delivery (Figure 1R4, see below). We noted that MC3 LNP failed to achieve detectable GFP signals in the liver at an mRNA dose of 0.05 mg/kg (Figure 1R4a). Since MC3 LNP is a positive control in our study, we used a moderate dose (0.25 mg/kg) in order to obtain detectable GFP signals for MC3 LNP.

Figure 1R4. *Ex vivo* fluorescence imaging of major organs. **a**, GFP expression in major organs at 4 h post-treatment of GFP mRNA-loaded LNPs at an mRNA dose of 0.05 mg/kg. **b**, GFP expression in major organs at 4 h post-treatment of GFP mRNA-loaded LNPs at an mRNA dose of 1 mg/kg. Mice were i.v. injected with GFP mRNA-loaded LNPs and *ex vivo* images were taken at 4 h post-treatment.

Reviewer #2 (Remarks to the Author):

General comments:

This is an interesting and systematic study reporting the effect of structure of branched lipidoids on the ability to delivery mRNA, Cas9/sgRNA and siRNA. However, the manuscript has a number of flaws that require the author's attention.

What determines the delivery efficiency is not only the chemical structure of the ionizable lipid, but also the colloidal carrier. From the drawn structure of the lipidoids, it makes sense to use the space shuttle analogy. However, the authors do not actually show that the intracellular degradation of the branched tails cause dissociation and release of the cargo, which makes the space shuttle analogy rather speculative and more a selling point of the paper than solid hypothesis-based research. The endosomal escape and mRNA release may rather be the result of a structural change in the nanoparticles caused by the charge reversal and the DOPE component of the nanoparticles. I suggest that the authors take the space shuttle analogy out, unless they have more solid experimental proof for their space shuttle hypothesis.

The authors are using the terms ionizable lipids and lipidoids alternately without defining them. It is an interesting point to discuss the nomenclature of lipid nanoparticles and the term lipidoid. It seems that the term "lipidoid" is most appropriate as these materials are lipid-like, and not truly lipids as defined previously (A combinatorial library of lipid-like materials for delivery of RNAi therapeutics | Nature Biotechnology) and also in this manuscript. However, they can also be termed as synthetic amino lipids (a term that Moderna also uses: A Novel Amino Lipid Series for mRNA Delivery: Improved Endosomal Escape and Sustained Pharmacology and Safety in Non-human Primates: Molecular Therapy (cell.com)).

The authors have chosen to use MC3/the Onpattro formulation for reference purposes, which was developed and optimized specifically for delivery of small interfering RNA to the liver. Although MC3 is used in general by many research groups as the 'benchmark' ionizable lipid OR lipidoid (in this context), it was not to be used as a benchmark for mRNA delivery as stated in a review by the pioneers in the field (Lipid Nanoparticle Systems for Enabling Gene Therapies: Molecular Therapy (cell.com)). For example, plasmid DNA delivery is greatly enhanced by DLin-K2-DMA instead of MC3-DMA (In vivo delivery of plasmid DNA by lipid nanoparticles: the influence of ionizable cationic lipids on organ-selective gene expression - Biomaterials Science (RSC Publishing)). Thus, in this case, if the delivery of plasmid DNA is to be considered, the benchmark would be KC2-DMA. Hence, MC3 is not the right benchmark lipid to use for mRNA delivery. Therefore, the authors should compare with the gold standard ionizable cationic lipids used in the COVID-19 vaccines, i.e. SM102 and/or ALC-0315, and compare with these lipids using the i.m. route of administration. For library 1, the authors should report the physicochemical properties of the

LNPs. It is unlikely that all structures are equally well suited for forming nanoparticles using the same LNP composition for all formulations, and it is relevant to report also how the lipidoid structure influences the colloidal stability of the LNPs, also in the presence of serum proteins. The failure of some of the formulation could be due to suboptimal compositions and not (only) the structure of the lipidoid.

For all concentrations, please convert ng to nmol.

For many experiments, n = 2. It is not possible to report a standard deviation, if n = 2. Please report mean values \pm standard deviations of at least triplicates.

Response: We thank the reviewer for this important feedback. We also thank the reviewer for their time and effort in helping us improve the manuscript. We were excited to hear that the reviewer felt that this is an interesting and systematic study reporting the effect of the structure of branched lipidoids on the ability to deliver mRNA, Cas9/sgRNA and siRNA. We have now corrected all mistakes and performed experiments based on the reviewer's suggestions, which have significantly improved our manuscript.

We agree with reviewer that it is unlikely that all structures are equally well suited for forming nanoparticles using the same LNP composition for all formulations, and the failure of some of the formulations could be due to suboptimal compositions and not (only) the structure of the lipidoid. However, it is not practical to optimize every LNP composition for each LNP and use the optimal formulation of each LNP to evaluate their performance. Instead, we used the same LNP composition and only switched the DB-lipidoid during LNP formulation, which we believe is reasonable in order to screen a large library of DB-lipidoids in a resource-effective manner. In previous studies, researchers also used the same LNP composition to screen and evaluate the performance of different lipidoids (*Nature Biotechnology*, 2019, 37: 1174-1185; *Nature Biotechnology*, 2023, 41: 1410-1415).

For the concentrations, as far as we know, researchers in both academia and industry prefer to use mass units instead of molar units in the field of mRNA delivery (*Nature Biotechnology*, 2019, 37: 1174-1185; *Nature Biotechnology*, 2023, 41: 1410-1415; *Molecular Therapy*, 2018, 26: 1509-1519). Therefore, we decided to keep “ng” while adding “nmol” as an additional unit in the legends of **Figures 2b, 3b, S4 and S9**.

We have now removed the space shuttle analogy, defined "lipidoid", compared 11-10-8 LNP with SM-102 LNP for mRNA delivery using the i.m. route of administration (**Figure S20**), reported the physicochemical properties of the LNPs in Library 1 (**Figure S3**), and included more replicates (n = 3-5) for several studies as suggested by the reviewer. Please read our detailed responses below. We hope that the reviewer enjoys our revised manuscript.

Specific comments:

Title:

The title is misleading, because it is not the lipid nanoparticles that are branched, but the ionizable lipids used to form the nanoparticles. In addition, potency is an intrinsic drug property, not a property of the transporter. The nanoparticle transporters should rather be described as efficient, because they are mediating a process. Please correct accordingly.

Response: We thank the reviewer for these suggestions, and apologize for this misleading title. We have now revised the title as follows: “***In situ* combinatorial synthesis of degradable branched lipidoids for systemic delivery of mRNA therapeutics and gene editors**”.

Abstract:

Line 34: Please rephrase “synthetic challenges”

Response: We thank the reviewer for this suggestion. We have now rephrased this as follows: “Particularly, degradable lipidoids containing extended alkyl branches have received tremendous attention due to their clinical success, yet their optimization and investigation are largely underappreciated due to their **laborious synthesis**.”

Line 38 and throughout the manuscripts: Numbers 1-9 are written as words (also page 3, line 107)

Response: We thank the reviewer for pointing out this issue. We have now corrected this issue throughout the manuscript.

Line 46: Replace “potent” with “efficient” (see comment to the title)

Response: We thank the reviewer for this suggestion. We have now revised it accordingly.

Introduction:

Page 2, line 49: To date the mRNA technology is only used in two approved products, i.e. the COVID-19 mRNA vaccines. Hence, it has only enable breakthroughs in the prevention of COVID-19. Please modify the text to reflect this.

Response: We thank the reviewer for this suggestion. We have now revised this in the manuscript (Page 2, line 53) as follows: “Messenger RNA (mRNA) technology **holds great promise** in the treatment and prevention of a variety of pathological conditions,

including cancers, infectious diseases, metabolic disorders, and congenital diseases^{1,2}. Indeed, mRNA-based therapeutics have achieved clinical success in vaccines, protein supplementation therapies, and gene editing therapies²⁻⁵. Specifically, two mRNA vaccines (i.e., mRNA-1273 and BNT162b2) have been approved by the Food and Drug Administration (FDA) for COVID-19 prevention. mRNA is a large, negatively charged and instable molecule, which needs a carrier for efficient intracellular delivery^{6,7}.”

Page 2, line 51: Again, the authors are overselling the mRNA technology by stating “tremendous clinical success” . Only two products are approved. Please modify the text accordingly.

Response: We thank the reviewer for this suggestion. It is true that only two mRNA products are approved by FDA, while many are in clinical trials. We have now revised it accordingly. Please read our above responses.

Page 2, line 57: The authors are using the terms ionizable lipids and lipidoids alternately without defining them. See comment above about the use of the terms ionizable lipids and lipidoids. For this manuscript, is there a distinction between ionizable lipids (or lipidoids) or should it be ionizable lipids (also termed as lipidoids synonymously)? Please define what lipidoids are, and make clear what the difference is between lipidoid and ionizable lipids.

Response: We thank the reviewer for this comment, and apologize for this confusion. We think ionizable lipid and lipidoid are the same in our scenario, and there is no difference between them. They both describe a lipid containing an ionizable amino head and two (or more) alkyl tails. According to the preference, different research papers or groups may use ionizable lipid or lipidoid alternatively. We have now revised this and made it clear in our manuscript (Page 2, line 62) as follows: “LNPs are typically comprised of lipidoids (also known as ionizable lipids), phospholipids, cholesterol (Chol) and polyethylene glycol (PEG)-conjugated lipids^{9, 11, 12}.”

Page 3, line 77: The authors have summed up the references for use of the branched tail intermediates coupled to functional groups, but it is also vital to cite the literature for studies related to the headgroups. Especially since the crucial amine 11 from this study was the same as amine 76 from one of the earliest studies (A combinatorial library of lipid-like materials for delivery of RNAi therapeutics | Nature Biotechnology (Figure 1). In addition, many of the amines in this manuscript correspond to the amines used in the above study. Amine 1 from the manuscript is amine 80, amine 6 is amine 10, and so on.

Response: We thank the reviewer for the great suggestion. We have now included this discussion (Page 6, line 228) and cited these papers as follows: “The majority of these amines were selected from previous publications^{32, 33}.”

Page 3, lines 100-101: MC3-based LNPs are clinically approved for siRNA, not mRNA. Please modify the text accordingly.

Response: We thank the reviewer for this important suggestion. We have now modified the text (Page 3, line 94) as follows: “Multiple DB-lipidoids were identified to form potent LNPs for *in vivo* mRNA delivery, which were comparable to or more efficient than the benchmark DLin-MC3-DMA (MC3) LNP that was approved for hepatic delivery of small interference RNA (siRNA).”

Results:

Page 4, line 117: Part of the structure is not degradable. Please modify the text to reflect this.

Response: We thank the reviewer for this suggestion. We have now mentioned this in the manuscript (Page 4, line 143) as follows: “In addition, epoxides with short alkyl chains (five variations between 6 and 14 carbons) were used as body tails to minimize the molecular weights (< 500 Da) of non-degradable metabolites (Figure 2a and S2), since previous studies have suggested that small-molecule metabolites tend to undergo rapid elimination^{16, 20}.”

Page 4, line 128: The authors argue that the crude lipidoids can be used directly for LNP formulation without additional purification, but the argumentation is not clear. If the two reactions are 80% efficient, the combined efficiency is $0.8 \times 0.8 = 64\%$. How can the authors be sure that impurities are not affecting the LNP formulation?

Response: We thank the reviewer for this important question. We previously claimed that “both reactions are highly efficient with >80% overall yield”. We intended to say that the total yield for this two-step reaction was above 80%. We have now made it clear in the manuscript (Page 4, line 128) as follows: “both reactions are highly efficient with an overall yield above 80%”.

In our study, we chose two representative DB-lipidoids (1-10-8 and 11-10-8) and compared their crude and purified products for *in vivo* mRNA delivery efficiency (Figure S6d and Figure 4b). The results showed that purified DB-lipidoids demonstrated comparable *in vivo* mRNA delivery efficiency to their crude ones, suggesting that impurities did not affect LNP formulation. It is worth mentioning that crude lipidoids generated from combinatorial chemistry are commonly used for LNP formulation and initial screening to accelerate lipidoid screening (*Nature*

Biotechnology, 2008, 26(5), 561-569; *Nature Biotechnology*, 2019, 37(10), 1174-1185). For example, Miao and colleagues synthesized 1080 lipidoids (yield typically >70%) based on an isocyanide-mediated 3-component reaction, and used for LNP formulation and *in vitro/vivo* initial screening without purification (*Nature Biotechnology*, 2019, 37(10), 1174-1185).

Page 6, line 152: MC3 was developed for siRNA delivery. Hence, “gene delivery” should be corrected to “siRNA delivery” .

Response: We thank the reviewer for this correction. We have now corrected this accordingly (Page 4, line 143).

Page 6, lines 152-155: The authors aim to minimize the size of the non-degradable parts “for accelerated physiological clearance” . Is that achieved?

Response: We thank the reviewer for this question, and apologize for overstating this claim. We have now modified this claim (Page 4, line 143) as follows: “In addition, epoxides with short alkyl chains (five variations between 6 and 14 carbons) were used as body tails to minimize the molecular weights (< 500 Da) of non-degradable metabolites (Figure 2a and S2), since previous studies have suggested that small-molecule metabolites tend to undergo rapid elimination^{16, 20}.”

We intended to reduce the molecular weights of non-degradable metabolites based on previous studies, which suggested that small-molecule, non-degradable metabolites could undergo rapid clearance and improve the biocompatibility of LNPs (*Molecular Therapy*, 2013, 21: 1570-1578; *Molecular Therapy*, 2018, 26: 1509-1519). For example, Alnylam incorporated biodegradable ester linkages into the hydrocarbon chain of MC3 lipid and developed L319 lipid, which reduced the size of the non-degradable fragment for accelerated elimination (*Molecular Therapy*, 2013, 21: 1570-1578). A similar strategy was also adopted for Moderna’s ionizable lipids, where biodegradable ester linkages are introduced into their tails to reduce the size of non-degradable fragment (*Molecular Therapy*, 2018, 26: 1509-1519).

Page 6, line 161: “25 SRB-LNPs” should be “25 SRB-LNP formulations” . Please provide the molar ratios of the LNP components.

Response: We thank the reviewer for these suggestions. We have now revised it accordingly (Page 4, line 151) as follows: “The resulting combinatorially synthesized 25 DB-lipidoids in Library 1 were formulated into 25 DB-LNP formulations by pipette mixing along with 1,2-dioleoyl-sn-glycero-3-phosphoethanolamine (DOPE), Chol, and 1,2-dimyristoyl-rac-glycero-3-methoxypolyethylene glycol-2000

(DMG-PEG). The weight ratio of DB-lipidoid/DOPE/Chol/DMG-PEG was fixed at 16/10/10/3 for initial screening.”

For initial screening, we fixed the weight ratio of DB-lipidoid/DOPE/Chol/DMG-PEG at 16/10/10/3, since it enabled us to quickly prepare all LNP formulations. We also provided more details about the molar ratios of the LNP components in the **Materials and Methods** section and **Table S2**.

Page 6, lines 183-4: The sentence “Since the total C number in the tail region is closely related to the hydrophobicity, it is reasonable to see the optimal one to be 18” . The authors should explain why it is reasonable, and this should be moved to the discussion.

Response: We thank the reviewer for this important feedback. The total carbon number in the tail region is closely related to the hydrophobicity of DB-lipidoids. We found the optimal total carbon number to be 18 (**Figure 2g**). Increased or decreased total carbon number generally led to reduced transfection efficiency, presumably due to the non-optimal hydrophobicity of tails. Considering that many natural (e.g., phospholipids) and synthetic lipids (e.g., DOTAP and MC3) contain 18-carbon tails, we were not surprised to determine the optimal total carbon number to be 18.

We have now explained this in the Discussion (Page 13, line 447) as follows: “It is worth mentioning that since the total carbon number in the tail region is closely related to the hydrophobicity of DB-lipidoids, it is reasonable that the optimal one was determined to be 18, considering that many natural and synthetic lipids contain 18-carbon tails¹³.”

Page 7, line 198: “mRNA delivery potency” should be corrected to “mRNA delivery efficiency”

Response: We thank the reviewer for this correction. We have now corrected this accordingly.

Page 7, line 205: “Ingredient” should be “agent” or “excipient” .

Response: We thank the reviewer for this correction. We have now corrected this accordingly.

Page 7, lines 209-211: The authors should show that intracellular branch tail degradation causes dissociation and mRNA release, and not just speculate. This

is the basis for the space shuttle analogy.

Response: We thank the reviewer for this suggestion, and apologize for overstating this claim. We agree with the reviewer's claim "The endosomal escape and mRNA release may rather be the result of a structural change in the nanoparticles caused by the charge reversal and the DOPE component of the nanoparticles." Considering many non-degradable benchmark ionizable lipids (e.g., C12-200 and cKK-E12) can achieve efficient mRNA delivery, the degradation of the 11-10-8 lipidoid may not be the major driving force for the LNP disassembly and mRNA release.

We have now taken the space shuttle analogy out based on the reviewer's and editor's suggestion, and revised the sentence (Page 6, line 220) as follows: "Together, these results suggest that both branch tails are required for the potency of DB-lipidoid and they can be detached following degradation." We have also updated **Figure 1**.

Figure 1. Construction of DB-lipidoids and DB-LNP-mediated mRNA delivery. (a) A scheme describing the tandem and *in situ* combinatorial synthesis of DB-lipidoids based on a one-pot, two-step, 3-CR. An amine reacts with alkyl epoxide (body tail) and the resulting aminoalcohol lipidoid further reacts with acyl chloride (branch tail) *in situ* to afford DB-lipidoid. (b) A scheme describing LNP formulation. The ethanol solution containing DB-lipidoid, phospholipid, PEG-lipid, and cholesterol is rapidly mixed with the acidic aqueous solution containing mRNA to formulate DB-LNP. (c) A scheme describing DB-LNP-mediated hepatic mRNA delivery. Intravenously (i.v.) administered DB-LNP is taken up by liver cells. mRNA is translated into protein, and DB-lipidoid undergoes degradation.

Page 7, line 217: Do not start a sentence with a number.

Response: We thank the reviewer for this suggestion. We have revised this sentence

(Page 6, line 227) as follows: “In total 20 chemically diverse amines were tested, including monoamines, diamines, polyamines, and hydrazines (Figure S8).”

Page 9, line 251: Is the 5-fold difference claimed by the authors statistically significant?

Response: We thank the reviewer for this question. It is statistically significant. We have now updated Figure 3c to reflect this.

Page 10, line 295: Again, is the 7-fold difference statistically significant?

Response: We thank the reviewer for this question. It is statistically significant. We have now updated Figure 4b to reflect this.

Page 10, line 298: There is a typo in physiochemical, which should be physicochemical.

Response: We thank the reviewer for this correction. We have revised it accordingly.

Page 10, line 304: Avoid non-scientific and relative words like “good” .

Response: We thank the reviewer for this suggestion. We have revised it in the manuscript (Page 10, line 318) as follows: “Due to its ionization ability, 11-10-8 LNP induced minimal hemolysis at pH 7.4, but increased hemolysis at pH 6.0 (Figure S14).”

Page 12, lines 336-338: Although the route of administration was kept constant (i.v.) by comparing to MC3, it would have been a fair comparison, if the 11-10-8 LNP was administered subcutaneously or intramuscularly and compared head-to-head to Lipid 5 or Lipid H (SM-102), which is relevant for mRNA delivery.

Response: We thank the reviewer for this important suggestion. We have now compared the mRNA delivery efficiency of 11-10-8 LNP and SM-102 LNP and included these data in Figure S19. The results showed that our 11-10-8 LNP-mediated stronger mRNA transfection after i.m. injection compared to SM-102 LNP. We have now updated the discussion in the manuscript (Page 13, line 416) as follows: “Apart from systemic delivery, we further showed that 11-10-8 LNP mediated strong intramuscular (i.m.) mRNA delivery (Figure S20), which

outperformed the benchmark SM-102 LNP that has been approved by FDA for mRNA vaccine delivery⁸. These results demonstrate the potential of 11-10-8 LNP for local mRNA delivery and mRNA vaccine development.”

Fig. S20 *In vivo* mLuc expression after i.m. injection of LNPs. Mice were i.m. injected with mLuc-loaded LNPs at an mRNA dose of 0.1 mg/kg. Images were taken at 4 h post-treatment. Data are presented as mean \pm SD (n = 3).

Page 12, line 345-347: Please adjust the number of significant figures. Commas should be corrected to full stops.

Response: We thank the reviewer for this suggestion. We have now repeated this experiment in high fat diet-induced obese mice (suggested by **Reviewer #1**) and updated these results in **Figure 6a**. We have also updated the discussion in the manuscript (Page 11, line 371) as follows: “The expression of FGF21 peaked (58770 ± 1087 pg/mL) at 12 h post-administration of FGF21 mRNA-loaded 11-10-8 LNP, which was five-fold higher than that in FGF21 mRNA-loaded MC3 LNP-treated mice (11764 ± 1064 pg/mL, **Figure 6a**). Moreover, the area under curve (AUC) – a pharmacokinetic metric of therapeutic exposure – of FGF21 was 4.6-fold greater in 11-10-8 LNP-treated mice than that in MC3 LNP-treated mice (**Table S5**).”

Page 12, line 365: The authors claim equal potency. This should be confirmed statistically

Response: We thank the reviewer for this suggestion. We have now performed statistical analysis (**Figure S19a**) and confirmed their comparable potency.

Figure S19. *In vivo* LNP-mediated TTR siRNA delivery. (a) Dose-dependent TTR silencing ($n = 3$). Mice were i.v. injected with TTR siRNA-loaded LNPs at different doses. Serum was collected on day 3 for ELISA analysis of serum TTR. (b) Duration of TTR silencing ($n = 3$). Mice were i.v. injected with TTR siRNA-loaded LNPs at a dose of 1 mg/kg. Serum was collected at indicated time points for ELISA analysis of serum TTR. Data are presented as mean \pm SD.

Page 13, last paragraph: This section should be conclusions, not just a summary.

Response: We thank the reviewer for this important feedback. We have now revised it (Page 14, line 480) as follows: “In conclusion, we devised a novel construction method that enables one-pot, high-throughput, and cost-efficient synthesis of DB-lipidoids. We identified multiple potent DB-lipidoids through combinatorial synthesis and screening of two libraries, and summarized key structural criteria governing the potency that can be used to predict the performance of unidentified analogs and guide the discovery of potent ones. Our lead DB-lipidoid outperformed the benchmark lipid MC3 in terms of hepatic mRNA delivery, demonstrating great potential for mRNA-based protein supplementation therapy and gene editing therapy. Overall, our new construction method lowers the threshold for synthesizing branched lipidoids, and this study lays a foundation for the further development and application of branched lipidoids for mRNA delivery.”

Materials and methods

Please include city and country of the headquarters of all vendors.

Response: We thank the reviewer for this suggestion. We have now revised them accordingly.

Page 14, line 454: Please explain “siRNA pool” . Is it a 1:1:1 molar ratio of the three different siRNAs? The sequences should be provided.

Response: We thank the reviewer for these comments. We pooled three different TTR siRNAs at a 1:1:1 molar ratio. We have now provided this information in the **Materials and Methods** (Page 18, line 658) as follows: “Three TTR siRNAs were pooled at a 1:1:1 molar ratio.”

These siRNAs were commercially available as we described in the **Materials** section (Page 15, line 505): “TTR siRNAs (#NM_013697, siRNA IDs: SASI_Mm01_00076059, SASI_Mm01_00076060 and SASI_Mm01_00076061) were purchased from Sigma Aldrich (Burlington, Massachusetts, USA).” However, Sigma Aldrich did not disclose their sequences (<https://www.sigmaaldrich.com/US/en/semi-configurators/sirna?term=TTR>) on their website and unfortunately they refused to tell us about the sequences.

Page 14, line 489: Is this ratio weight ratio or molar ratio? Please provide the lipid molar ratio (in percent) to enable direct comparison with the MC3 formulation + lipid:mRNA weight ratio.

Response: We thank the reviewer for these comments. This is the weight ratio. For initial screening, we fixed the weight ratio of DB-lipidoid/DOPE/Chol/DMG-PEG at 16/10/10/3 and the weight ratio of lipidoid/mRNA at 10:1, since it enabled us to quickly prepare all LNP formulations. A similar strategy was used by others (*PNAS*, 2016, 113: 2868-2873; *Angew. Chem. Int. Ed.* 2020, 59, 20083-20089). In this case, the molar ratio of lipids would change based on the molecular weight of lipidoid. We further optimized the LNP formulation once we identified the lead DB-lipidoid 11-10-8 (**Figure 4b**). **Table S2** shows the lipid molar ratio (in percent) of our lead 11-10-8 LNP and MC3 LNP. F1 is the formulation used for initial screening, while F5 is the optimized formulation.

Table S2. LNP formulation tested and their sources.

Formulation	Recipe	Molar ratio	Weight ratio	Source
F1	11-10-8*/DOPE/Chol/DMG-PEG	36.8:21:40.4:1.8	16:10:10:3	/
F2	11-10-8/DOPE/Chol/DMG-PEG	36.8:21:40.4:1.8	16:10:10:3	/
F3	11-10-8/DOPE/Chol/DMG-PEG	30.6:17.5:50.4:1.5	16:10:15:3	In house
F4	11-10-8/DOPE/Chol/DMG-PEG	35:16:46.5:2.5	16:8:12:4.5	Ref. ⁶
F5	11-10-8/DOPE/Chol/DMG-PEG	40:10:48.5:1.5	16:4.4:11:2.4	Ref. ⁷
F6	11-10-8/DSPC/Chol/DMG-PEG	50:10:38.5:1.5	16:3.7:7:1.9	Ref. ⁸
MC3 LNP	MC3/DSPC/Chol/DMG-PEG	50:10:38.5:1.5	16:3.7:7:1.9	

11-10-8* indicates crude 11-10-8. The weight ratio of lipidoid:RNA was kept at 10:1 during

LNP formulation.

Page 14, line 495. Is this the molar ratio? Please state that.

Response: We thank the reviewer for this question. This is the molar ratio. We have now improved it (Page 15, line 545) as follows: “To optimize the formulation of 11-10-8 DB-LNP, various LNPs formulated by microfluidic mixing were tested *in vivo*, and the optimal one with a molar ratio of 11-10-8/DOPE/Chol/DMG-PEG at 40:10:48.8:1.5 was chosen for subsequent studies.”

Figures:

Figure 1: This figure is rather speculative and indeed misleading, because the authors are not showing in the manuscript that nanoparticle disassembly is caused by the cleavage of the ester bonds in the ionizable lipids. The endosomal escape and mRNA release may rather be the result of a structural change in the nanoparticles caused by the charge reversal and the DOPE component of the nanoparticles. I suggest that the authors take the figure out, unless they have more solid experimental proof for their space shuttle hypothesis.

Response: We thank the reviewer for these comments. We completely agree with the reviewer. Considering many non-degradable benchmark ionizable lipids (e.g., C12-200 and cKK-E12) can achieve efficient siRNA and mRNA delivery, the cleavage of the ester bonds for our 11-10-8 lipidoid may not be the major driving force for the LNP disassembly and mRNA release.

We have now taken the space shuttle analogy out and updated **Figure 1**.

Figure 1. Construction of DB-lipidoids and DB-LNP-mediated mRNA delivery. (a) A scheme describing the tandem and *in situ* combinatorial synthesis of DB-lipidoids based on a one-pot, two-step, 3-CR. An amine reacts with alkyl epoxide (body tail) and the resulting aminoalcohol lipidoid further reacts with acyl chloride (branch tail) *in situ* to afford DB-lipidoid. (b) A scheme describing LNP formulation. The ethanol solution containing DB-lipidoid, phospholipid, PEG-lipid, and cholesterol is rapidly mixed with the acidic aqueous solution containing mRNA to formulate DB-LNP. (c) A scheme describing DB-LNP-mediated hepatic mRNA delivery. Intravenously (i.v.) administered DB-LNP is taken up by liver cells. mRNA is translated into protein, and DB-lipidoid undergoes degradation.

Figure 2: Please include mean values for triplicates in c) and d), and state in the legend that mean values are reported. For c), please state the cell type that is used.

Response: We thank the reviewer for these suggestions. We have now included more replicates and updated **Figure 2** and its legend accordingly.

Figure 2. Optimization of the tail region and screening of Library 1. (a) A workflow for the synthesis and evaluation of Library 1. (b) *In vitro* mLuc expression shown in a heat map ($n = 3$). HepG2 cells were treated with mLuc-loaded LNPs at an mRNA dose of 15 ng/well for 24 h. RLU, relative light unit. Data are presented as mean. (c) *In vivo* mLuc expression shown in a heat map ($n = 3$). Mice were i.v. injected with mLuc-loaded LNPs at an mRNA dose of 0.1 mg/kg. Bioluminescence imaging (BLI) was performed at 4 h post-treatment and total flux was quantified. Data are presented as mean. (d) Correlation between *in vitro* and *in vivo* results of DB-lipidoids. The black dashed line indicates 10,000 RLU *in vitro*. The blue dashed line indicates the performance of MC3 LNP *in vivo*. (e) Body tail-activity relationship. (f) Branch tail-activity relationship. (g) Total carbon number-activity relationship. (h) Total carbon number-symmetry-activity relationship. The grey shadows indicate background levels.

Figure 3: What is shown in b) and c)? Mean values? Please state that in the legend. It is not possible to have standard deviation when $n = 2$. The *in vivo* luminescence background in this graph is 10^7 , while that in Figures 4a and 4b is 10^5 . This is a huge difference in terms of the values, because the total flux for the MC3-based formulation is practically the same. It would be great to know the

author's input on this.

Response: We thank the reviewer for these questions and suggestions. We have now included more replicates and updated **Figure 3** and its legend accordingly. For *in vivo* luminescence background, it is 10^7 p/s throughout this study. We have now updated **Figure 4** and its legend accordingly.

Figure 3. Optimization of the headgroup and screening of Library 2. (a) A workflow for the synthesis and evaluation of Library 2. (b) *In vitro* mLuc expression (n = 3). HepG2 cells were treated with mLuc-loaded LNPs at an mRNA dose of 15 ng/well for 24 h. Data are presented as mean ± SD. (c) *In vivo* mLuc expression (n = 3). Mice were i.v. injected with mLuc-loaded LNPs at an mRNA dose of 0.1 mg/kg. BLI was performed at 4 h post-treatment and total flux was quantified. Efficacious DB-lipidoids and their amines are highlighted in red. The grey shadow indicates background level. Data are presented as mean ± SD. (d) Correlation between *in vitro* and *in vivo* results of DB-lipidoids. The black dashed line indicates 10,000 RLU *in vitro*. The blue dashed line indicates the performance of MC3 LNP *in vivo*. (e) Structures of efficacious amines and DB-lipidoids. The chemical structure of lead DB-lipidoid 11-10-8 is shown. 11-10-8 demonstrates a total carbon number of 18, a

symmetry of 1, and a packing parameter (P) of 4.1.

Figure 4. Prediction, optimization, and characterization of DB-LNPs. (a) Prediction and verification of *in vivo* performance for unidentified DB-lipidoids ($n = 3$). Mice were i.v. injected with mLuc-loaded LNPs at an mRNA dose of 0.1 mg/kg. BLI was performed at 4 h post-treatment and total flux was quantified. The red dashed line indicates the performance of 11-10-8 LNP, while the blue dashed line indicates the performance of MC3 LNP. The grey shadow indicates background level. (b) Optimization of 11-10-8 LNP formulation ($n = 3$). Mice were injected i.v. with mLuc-loaded LNPs at an mRNA dose of 0.1 mg/kg. BLI was performed at 4 h post-treatment and total flux was quantified. The grey shadow indicates background level. (c) A representative cryo-EM image of 11-10-8 LNP. Scale bar = 50 nm. (d) Physicochemical properties of 11-10-8 LNP ($n = 3$). (e) Inhibition of 11-10-8 LNP uptake by various endocytic inhibitors ($n = 3$). Amiloride is an inhibitor of macropinocytosis; Chlorpromazine is an inhibitor of clathrin-mediated endocytosis; Genistein is an inhibitor of caveolae-mediated endocytosis; Methyl- β -cyclodextrin (M β -CD) is an inhibitor of lipid raft-mediated endocytosis. Data are presented as mean \pm SD.

Figure 4: For the *in vivo* experiments, $n = 2$, which is too little for meaningful data, considering the biological variation. Please include more replicates for proper statistics. There is a typo in zet potential. In the legend, there is a typo in physiochemical, which should be physicochemical. In the same line “parameters”

should be “properties”. “Lipid rafter-mediated” should be “lipid raft-mediated”.

Response: We thank the reviewer for these suggestions and corrections. We have now performed the *in vivo* experiments using extra mice and included more replicates for proper statistics (n = 3). Moreover, we have corrected all typos found by the reviewer. We have now updated **Figure 4** and its legend accordingly.

Figure 4. Prediction, optimization, and characterization of DB-LNPs. (a) Prediction and verification of *in vivo* performance for unidentified DB-lipidoids (n = 3). Mice were i.v. injected with mLuc-loaded LNPs at an mRNA dose of 0.1 mg/kg. BLI was performed at 4 h post-treatment and total flux was quantified. The red dashed line indicates the performance of 11-10-8 LNP, while the blue dashed line indicates the performance of MC3 LNP. The grey shadow indicates background level. (b) Optimization of 11-10-8 LNP formulation (n = 3). Mice were injected i.v. with mLuc-loaded LNPs at an mRNA dose of 0.1 mg/kg. BLI was performed at 4 h post-treatment and total flux was quantified. The grey shadow indicates background level. (c) A representative cryo-EM image of 11-10-8 LNP. Scale bar = 50 nm. (d) Physicochemical properties of 11-10-8 LNP (n = 3). (e) Inhibition of 11-10-8 LNP uptake by various endocytic inhibitors (n = 3). Amiloride is an inhibitor of macropinocytosis; Chlorpromazine is an inhibitor of clathrin-mediated endocytosis; Genistein is an inhibitor of caveolae-mediated endocytosis; Methyl- β -cyclodextrin (M β -CD) is an inhibitor of lipid raft-mediated endocytosis. Data are presented as mean \pm SD.

Figure 5: Although $n = 3$ is considered as the minimum requirement for applying statistics, based on the deviations for MC3 in Figure 5d, it is appropriate to include more number of samples for analysis. For the graphs in c) and e), data points should not be connected with full lines. Full lines are reserved for mathematical fits or continuous data sets. More data points should be included.

Response: We thank the reviewer for these suggestions and corrections. We have now performed *in vivo* TTR knockout using more mice in updated **Figure 5c,d** ($n = 5$). In line with our previous results, 11-10-8 LNP achieved greater TTR gene editing efficiency (30% vs 7%) and serum TTR reduction (50% vs 17%) than MC3 LNP.

For the graphs in updated **Figure 5d** and **Figure 6a**, we have now connected data points with dashed lines and included more data points.

Figure 5. DB-LNP-mediated hepatic delivery of mRNA-based gene editors. (a) *Ex vivo* BLI of major organs from treated mice and their quantification ($n = 3$). Mice were i.v. injected with mLuc-loaded LNPs at an mRNA dose of 0.1 mg/kg. Images were taken at 4 h post-treatment. (b) *Ex vivo* fluorescence imaging of major organs

from treated mice and their quantification (n = 3). Mice were i.v. injected with GFP mRNA-loaded LNPs at an mRNA dose of 0.25 mg/kg. Images were taken at 4 h post-treatment. (c, d) LNP-mediated Cas9 mRNA/TTR sgRNA co-delivery and gene editing. Mice were i.v. injected with LNPs co-delivering Cas9 mRNA/TTR sgRNA (4:1, wt:wt) at a total RNA dose of 1 mg/kg. Mice were euthanized on day 7, and DNA was extracted from the liver to determine on-target indel frequency by next-generation sequencing (c, n = 5). Serum was collected at the indicated time points for ELISA analysis of TTR (d, n = 5). Data are presented as mean \pm SD.

Figure 6. DB-LNP-mediated hepatic delivery of FGF21 mRNA. (a) LNP-mediated FGF21 mRNA delivery (n = 4). Male mice were fed a HFD for three weeks to induce obesity (body weight ~30g) and fatty liver. These obese mice were i.v. injected with FGF21 mRNA-loaded LNPs at an mRNA dose of 0.25 mg/kg. Serum was collected at the indicated time points for ELISA analysis of FGF21. (b) A scheme of FGF21 mRNA therapy in HFD-induced obese mice. Obese mice were i.v. injected with various LNP formulations at an mRNA dose of 0.25 mg/kg every other day for three doses. Male mice fed a normal chow diet (NCD) was used a control group. (c) Body weight growth curve (n = 4). (d) Body weight on Day 6 (n = 4). (e) Liver weight (n = 4). (f) Representative images of liver histological examinations (H&E staining). Scale bars: 100 μ m. (g) Liver steatosis score (n = 12). A total of 12 images for each group (three random fields for each liver section, four mice per group) were analyzed semi-quantitatively for liver steatosis. Data are presented as mean \pm SD.

Supplementary data:

Figure S1: Please include the structure of MC3 for comparison. What is the biophysical rationale for the design of these structures?

Response: We thank the reviewer for this question. We have now included the structure of MC3 in updated **Figure S2**. We have discussed the biophysical rationale for the design of Library 1 in the section **2.2 Optimizing the tail region of DB-lipidoids** as follows: “Therefore, in the first library (Library 1), the headgroup was kept constant as amine 1 (i.e., 3-(dimethylamino)-1-propylamine) and the tail regions – both body tail and branch tail – were varied (**Figures 2a, S2 and Table S1**). Amine 1 was chosen based on the studies of MC3 lipidoid, which suggest that the dimethylamino moiety with a spacer of three methylene units is effective for siRNA delivery^{29, 30}. In addition, epoxides with short alkyl chains (five variations between 6 and 14 carbons) were used as body tails to minimize the molecular weights (< 500 Da) of non-degradable metabolites (**Figures 2a and S2**), since previous studies have suggested that small-molecule metabolites tend to undergo rapid elimination^{16, 20}. Correspondingly, acyl chlorides with short alkyl chains (five variations between 6 and 14 carbons) were selected as branch tails.”

Figure S2. Chemical structures of DB-lipidoids in Library 1, aminoalcohol lipidoids and DLin-MC3-DMA (MC3).

Figure S2: Legend of x-axis: What does “experiment no” refer to? Which structure? It is not possible to have standard deviation when $n = 2$. The dose should be given in nmol, not ng. The same comments go for Figure S7.

Response: We thank the reviewer for these questions and suggestions. We have

included more replicates and updated **Figures S4** and **S9** and their legends accordingly. Since researchers in both academia and industry prefer to use mass units instead of molar units in the field of mRNA delivery (*Nature Biotechnology*, 2019, 37: 1174-1185; *Nature Biotechnology*, 2023, 41: 1410-1415; *Molecular Therapy*, 2018, 26: 1509-1519). Therefore, we decided to keep “ng” while adding “nmol” as an additional unit in the legends.

Figure S4. Cell viability of DB-LNPs in Library 1 and other LNPs. HepG2 cells were treated with various LNPs at a dose of 15 ng mRNA/well (2.4×10^{-5} nmol mRNA/well) for 24 h. No major cytotoxicity was induced by LNPs. The dashed line indicates 80% cell viability. Data are presented as mean \pm SD (n = 3).

Figure S8. Cell viability of DB-LNPs in Library 2. HepG2 cells were treated with various LNPs at a dose of 15 ng mRNA/well (2.4×10^{-5} nmol mRNA/well) for 24 h. No major cytotoxicity was induced by DB-LNPs. The dashed line indicates 80% cell viability. Data are presented as mean \pm SD (n = 3).

Figure S11: The polynomial fit is wrong. Data should be fitted to a sigmoidal curve.

Response: We thank the reviewer for this important comment. We have now revised it accordingly and updated **Figure S13**. The apparent pK_a of 11-10-8 LNP was determined to be 6.22.

Figure S13. TNS assay was used to determine the apparent pK_a of 11-10-8 LNP. TNS fluorescence signal corresponds to ionization. pK_a is calculated as the pH corresponding to half of the maximum TNS fluorescence value.

Figure S16: Three data points are not enough for proper determination of ED50 values. Please include more data points.

Response: We thank the reviewer for this suggestion. We have now included more data points and re-determined ED50. The ED50 for 11-10-8 LNP was 0.029 mg/kg, which was comparable to MC3 LNP (0.03 mg/kg, **Figure S19a**). We have now updated this discussion in the manuscript (Page 13, line 409) as follows: “Interestingly, when TTR siRNA was delivered, 11-10-8 LNP and MC3 LNP showed comparable potency with the similar median effective dose (ED50, 0.029 mg/kg vs 0.030 mg/kg, **Figure S19a**).”

Figure S19. *In vivo* LNP-mediated TTR siRNA delivery. (a) Dose-dependent TTR silencing ($n = 3$). Mice were i.v. injected with TTR siRNA-loaded LNPs at different doses. Serum was collected on day 3 for ELISA analysis of serum TTR. (b) Duration of TTR silencing ($n = 3$). Mice were i.v. injected with TTR siRNA-loaded LNPs at a dose of 1 mg/kg. Serum was collected at indicated time points for ELISA analysis of serum TTR. Data are presented as mean \pm SD.

Table S2: Are the same formulations tested for siRNA, Cas9/sgRNA and mRNA? What are the N/P ratios (or weight ratios of nucleic acid cargo)?

Response: We thank the reviewer for these questions. Once we obtained the optimal 11-10-8 LNP formulation (F5), we used it in our following experiments to deliver siRNA, Cas9/sgRNA and mRNA. We kept the weight ratio of lipidoid:RNA at 10:1 throughout our study. We have now updated **Table S2**.

Table S2. LNP formulation tested and their sources.

Formulation	Recipe	Molar ratio	Weight ratio	Source
F1	11-10-8*/DOPE/Chol/DMG-PEG	36.8:21:40.4:1.8	16:10:10:3	/
F2	11-10-8/DOPE/Chol/DMG-PEG	36.8:21:40.4:1.8	16:10:10:3	/
F3	11-10-8/DOPE/Chol/DMG-PEG	30.6:17.5:50.4:1.5	16:10:15:3	In house
F4	11-10-8/DOPE/Chol/DMG-PEG	35:16:46.5:2.5	16:8:12:4.5	Ref. ⁶
F5	11-10-8/DOPE/Chol/DMG-PEG	40:10:48.5:1.5	16:4.4:11:2.4	Ref. ⁷
F6	11-10-8/DSPC/Chol/DMG-PEG	50:10:38.5:1.5	16:3.7:7:1.9	Ref. ⁸
MC3 LNP	MC3/DSPC/Chol/DMG-PEG	50:10:38.5:1.5	16:3.7:7:1.9	

11-10-8* indicates crude 11-10-8. The weight ratio of lipidoid:RNA was kept at 10:1 during LNP formulation.

Table S4: The plasma concentration profile has only three data point, of which one point is very low. Therefore, there are not enough data points on this curve to justify determination of the AUC values given in Table S4.

Response: We thank the reviewer for this important comment. We have now repeated this experiment in high fat diet-induced obese mice as suggested by **Reviewer #1** and included more data points (**Figure 6a**). We have now updated these results in **Table S5**.

Table S5. Serum FGF21 protein expression in **obese** mice during the time interval 0–48 h.

LNP	11-10-8 LNP	MC3 LNP
AUC (pg·h/mL)	1104069 ± 12245	240378 ± 7767

AUC, area under curve. **Obese mice** were i.v. injected with FGF21 mRNA-loaded LNPs at an mRNA dose of 0.25 mg/kg. Serum was collected at **0, 3, 6, 12, 24 and 48 h** post-injection and analyzed by ELISA. Data are presented as mean ± SD (n = 4).

Reviewer #3 (Remarks to the Author):

I co-reviewed this manuscript with one of the reviewers who provided the listed reports as part of the Nature Communications initiative to facilitate training in peer review and appropriate recognition for co-reviewers.

Response: We thank the reviewer for co-reviewing this manuscript, and hope that the reviewer enjoys our revised manuscript.

Reviewer #4 (Remarks to the Author):

Focus of this manuscript is majorly chemistry and optimization. Seems the content is best focus for chemistry journal like ACIE, Chem Sci, JACS, JCR.

Response: We thank the reviewer for this comment. We also thank the reviewer for their time and effort in helping us improve the manuscript.

While the focus of the previous version of our manuscript is majorly chemistry and optimization, the relevance of our lead 11-10-8 LNP towards clinical applications is demonstrated by the delivery of FGF21 mRNA, Cas9 mRNA/TTR sgRNA and TTR siRNA. With approximately five-fold higher therapeutic protein expression and genome editing efficiency compared to the benchmark MC3 LNP and equivalent knockdown efficiency, our 11-10-8 LNP demonstrates translational potential.

In the revised version of our manuscript, we further ascertain the relevance of 11-10-8 LNP towards clinical application by performing a therapeutic study using FGF21 mRNA to treat obesity and fatty liver. In a diet-induced obese mouse model, we demonstrated that systemic delivery of FGF21 mRNA by 11-10-8 LNP exhibited superior weight-reducing and lipid-lowering effects compared to MC3 LNP, resulting in significantly alleviated obesity and fatty liver (**Figure 6**).

Therefore, we strongly feel our work involving chemical, biological and biomedical sciences is suitable for *Nature Communications*.

Figure 6. DB-LNP-mediated hepatic delivery of FGF21 mRNA. (a) LNP-mediated FGF21 mRNA delivery (n = 4). Male mice were fed a HFD for three weeks to induce

obesity (body weight ~30g) and fatty liver. These obese mice were i.v. injected with FGF21 mRNA-loaded LNPs at an mRNA dose of 0.25 mg/kg. Serum was collected at the indicated time points for ELISA analysis of FGF21. (b) A scheme of FGF21 mRNA therapy in HFD-induced obese mice. Obese mice were i.v. injected with various LNP formulations at an mRNA dose of 0.25 mg/kg every other day for three doses. Male mice fed a normal chow diet (NCD) was used a control group. (c) Body weight growth curve (n = 4). (d) Body weight on Day 6 (n = 4). (e) Liver weight (n = 4). (f) Representative images of liver histological examinations (H&E staining). Scale bars: 100 μ m. (g) Liver steatosis score (n = 12). A total of 12 images for each group (three random fields for each liver section, four mice per group) were analyzed semi-quantitatively for liver steatosis. Data are presented as mean \pm SD.

Authors may explain better how the molecule is like space shuttle. Chemical design are popular amine-epoxide ring opened lipidoids with hydroxyl position modified to add more hydrophobic tails. The carton look like LNP with rocket ship conjugated to the surface like targeting ligand, but actual structure is regular LNP structure. This is little confusing.

Response: We thank the reviewer for this important feedback and apologize for the overstated analogy. We have now removed the space shuttle analogy throughout the manuscript, which is also requested by **Reviewer #2** and the editor. We have now updated **Figure 1**.

Figure 1. Construction of DB-lipidoids and DB-LNP-mediated mRNA delivery. (a) A scheme describing the tandem and *in situ* combinatorial synthesis of DB-lipidoids based on a one-pot, two-step, 3-CR. An amine reacts with alkyl epoxide (body tail) and the resulting aminoalcohol lipidoid further reacts with acyl chloride (branch tail) *in situ* to afford DB-lipidoid. (b) A scheme describing LNP formulation. The ethanol solution containing DB-lipidoid, phospholipid, PEG-lipid, and

cholesterol is rapidly mixed with the acidic aqueous solution containing mRNA to formulate DB-LNP. (c) A scheme describing DB-LNP-mediated hepatic mRNA delivery. Intravenously (i.v.) administered DB-LNP is taken up by liver cells. mRNA is translated into protein, and DB-lipidoid undergoes degradation.

Also other papers show role of branched tails in mRNA delivery already published. Hashiba et al. “Branching Ionizable Lipids Can Enhance the Stability, Fusogenicity, and Functional Delivery of mRNA” *Small Science*, Volume 3, 2200071, 2023. Hajj et al. “Branched-Tail Lipid Nanoparticles Potently Deliver mRNA In Vivo due to Enhanced Ionization at Endosomal pH” *Small*, 15, 1805097, 2019. Hajj et al. “A Potent Branched-Tail Lipid Nanoparticle Enables Multiplexed mRNA Delivery and Gene Editing In Vivo” *Nano Letters* 20, 5167, 2020. Sabnis et al. “A Novel Amino Lipid Series for mRNA Delivery: Improved Endosomal Escape and Sustained Pharmacology and Safety in Non-human Primates” *Molecular Therapy*, 26, 1509, 2018.

Response: We thank the reviewer for this comment. We cited and discussed most of them in our previous version of manuscript. Although the role of branched tails in mRNA delivery has been shown in previous publications, the deep understanding of how the branched structure (e.g., total carbon number and symmetry) influences *in vivo* performance is still limited. Moreover, the structures of branched lipidoids are very different (**Figure S1**), suggesting that optimal structures obtained in one kind of branched lipidoid may not be applicable to the other.

Currently, degradable lipidoids containing extended alkyl branches have received tremendous attention due to the clinical success of Moderna’s SM-102 and Acuitas’s ALC-0315. However, it is challenging to build a large and systematically-designed library of branched lipidoids with varying lengths of body tail and branch tail based on previous synthetic methods (**Figure S1**), making their optimization and investigation largely hampered. We sought to address this challenge. For the first time, we devised a tandem and *in situ* construction method for rapid, cost-efficient, and high-throughput synthesis of degradable branched lipidoids. This facile construction method avoids the use of branched intermediates and allows for precise and independent control of each structural parameter, including headgroup, body tail, branch tail, and symmetry. Compared to a previous strategy for CL4F m-n lipid synthesis (**Figure S1**), our construction method is more concise and flexible, enabling the generation of more structurally diverse branched lipidoids.

We have now summarized representative synthetic routes for degradable lipidoids with extended alkyl branches from the literature to demonstrate the superiority of our synthetic method (**Figure S1**).

Figure S1. Summary of representative synthetic routes for degradable lipidoids with extended alkyl branches from the literature. The synthetic routes of Moderna's Lipid 5¹, Acuitas's ALC-0315², AX4³, Genevant's Lipid-10⁴ and CL4F m-n lipids⁵ are shown. These lipidoids were synthesized based on two main steps: first, the preparation of a branched tail intermediate containing a functional group (highlighted in red); second, the connection of branched tail(s) to the headgroup. This method involves multiple synthetic steps and purifications with limited capacity (due to the lack of readily available branched intermediate) to generate a large library of degradable branched lipidoids.

REVIEWER COMMENTS

Reviewer #1 (Remarks to the Author):

Authors have addressed all of my concerns. I am satisfied with improved data in revised version of this manuscript.

Additional Review of comments of Reviewer 4 by Reviewer 1:

In their revised manuscript, authors have addressed all the concerns of reviewer 4 adequately. They have now provided sufficient preclinical therapeutic data on delivery of FGF21 mRNA, Cas9mRNA/TTR sgRNA and TTR siRNA using 11-10-8 LNP. Of note, five-fold higher protein expression and the potential treatment of obesity and lipid-lowering effects were demonstrated using 11-10-8 LNP.

Additionally, following the reviewer's suggestions, authors have now omitted a few statements (such as space shuttle analogy) that may have appeared exaggeration to some extent in the first version of manuscript. This has certainly improved the clarity of the revised manuscript.

On a minor note, authors have also now discussed all references that was mentioned by the reviewer. Taken together, I believe that authors have now addressed all comments of this reviewer. Hence, I recommend publication of this manuscript in Nature Communications.

Reviewer #2 (Remarks to the Author):

General comments:

The authors have done a good job and addressed most of the comments. However, there are still some flaws that require attention:

The authors should add a sentence or two to the manuscript stating that not all structures may be equally suited for forming nanoparticles, and that some formulations may be suboptimal. Indeed, the physicochemical properties (the encapsulation efficiency in particular) presented in Figure S3 actually suggests that several formulations are suboptimal (see below).

In supplementary Figure S3, the authors have now reported the physicochemical properties of the LNPs, which is good. A red heat map is used, but it is actually impossible to read the numbers in the dark red fields. In addition, standard deviations and statistics are missing. I suggest that the authors use a table format instead and also report the standard deviation of the numbers. In addition, please also adjust the significant figures. The rule is that the standard deviation provides a measurement of experimental uncertainty and should almost always be rounded to one significant figure. The only exception is when the uncertainty (if written in scientific notation) has a leading digit of 1, then a second digit should be kept. It is clear from the numbers that the mRNA encapsulation efficiency is very low for several of the formulations, and the numbers are in general quite low. It is preferable to have encapsulation efficiencies over 90% to be able to compare different formulations at equal mRNA loading.

When the encapsulation efficiency is low, more lipid is required to achieve the same mRNA dose, and that results in more toxicity. Eventually, high toxicity can cause shut down of protein synthesis.

Therefore, the data presented in Figure S4 is of little use, because the formulations are compared at very different lipid doses and at only one single mRNA concentration. Cell viability is determined at different doses, and the concentrations that results in 50% viability should be compared.

There is a mistake in the legend of Figure S3: The zeta potential is not determined by using dynamic light scattering, but by laser-Doppler electrophoresis. It is of little use to measure and compare the hydrodynamic size and the zetapotential in serum-containing medium by using the Zetasizer Nano. Both types of measurements are very dependent on the composition of the medium, and the presence of serum proteins interferes with the measurements. Other methods should be used to measure size and zetapotential in serum-containing medium, so I would suggest to take the data for the measurements in FBS-containing PBS out of the manuscript.

It was apparently not clear why it is preferable to provide the RNA concentration on a molar basis instead of a weight basis. The reason why RNA concentrations should be converted to molar concentration is to be able to compare the efficacy and safety of LNPs loaded with RNAs with different lengths, also between different studies. If you use a weight-based concentration, you can only perform a direct comparison of concentrations of RNAs with exactly the same length.

Specific comments:

Page 2, line 56: *i.e.* is written in italic

Page 2, line 57: Please call the COVID-19 mRNA vaccines by their names: Spikevax® and Comirnaty®

Page 2, line 58: "Instable" should be corrected to "unstable"

Page 3, line 96: "...LNP that was.." should be "...LNPs that were" or "...LNP formulation that was..."

Page 12, lines 372-374: Please adjust the significant figures. Again, the rule is that the standard deviation provides a measurement of experimental uncertainty and should almost always be rounded to one significant figure. The only exception is when the uncertainty (if written in scientific notation) has a leading digit of 1, then a second digit should be kept. Hence, 58770 ± 1087 pg/mL should be 58.8 ± 1.1 ng/mL, and 11764 ± 1064 pg/mL should be 11.8 ± 1.1 ng/mL.

Table S5: A table with only two numbers does not make sense. These two numbers can be written in the text or added to figure 6. Again, please adjust the significant figures as above.

Reviewer #3 (Remarks to the Author):

I co-reviewed this manuscript with one of the reviewers who provided the listed reports as part of the Nature Communications initiative to facilitate training in peer review and appropriate recognition for co-reviewers.

REVIEWERS' COMMENTS AND AUTHORS' ANSWERS

Note: Our responses (standard typeface) to reviewers' comments (bold); the yellow highlighted words and sentences have been added to the main text.

Reviewer #1 (Remarks to the Author):

Authors have addressed all of my concerns. I am satisfied with improved data in revised version of this manuscript.

Response: We thank the reviewer for this encouraging decision. We thank the reviewer again for their time and effort in helping us improve the manuscript.

Additional Review of comments of Reviewer #4 by Reviewer #1:

In their revised manuscript, authors have addressed all the concerns of reviewer 4 adequately. They have now provided sufficient preclinical therapeutic data on delivery of FGF21 mRNA, Cas9mRNA/TTR sgRNA and TTR siRNA using 11-10-8 LNP. Of note, five-fold higher protein expression and the potential treatment of obesity and lipid-lowering effects were demonstrated using 11-10-8 LNP.

Additionally, following the reviewer's suggestions, authors have now omitted a few statements (such as space shuttle analogy) that may have appeared exaggeration to some extent in the first version of manuscript. This has certainly improved the clarity of the revised manuscript.

On a minor note, authors have also now discussed all references that was mentioned by the reviewer.

Taken together, I believe that authors have now addressed all comments of this reviewer. Hence, I recommend publication of this manuscript in Nature Communications.

Response: We thank the reviewer for this encouraging decision. We thank the reviewer again for their time and effort in helping us improve the manuscript.

Reviewer #2 (Remarks to the Author):

General comments:

The authors have done a good job and addressed most of the comments. However, there are still some flaws that require attention:

Response: We thank the reviewer for these comments. We also thank the reviewer for their time and effort in helping us improve the manuscript. We were excited to hear that the reviewer felt that we have done a good job and addressed most of the comments. We have now corrected all mistakes and performed experiments based on the reviewer's suggestions, which have significantly improved our manuscript.

The authors should add a sentence or two to the manuscript stating that not all structures may be equally suited for forming nanoparticles, and that some formulations may be suboptimal. Indeed, the physicochemical properties (the encapsulation efficiency in particular) presented in Figure S3 actually suggests that several formulations are suboptimal (see below).

Response: We thank the reviewer for this suggestion. We have now added a sentence (Page 4, line 156) as follows: "It is should be noted that due to the divergent structures, not all lipidoids were equally suited for LNP formation and some LNP formulations might be suboptimal. In general, aminoalcohol lipidoids were inferior for mRNA encapsulation, while DB-lipidoids showed enhanced capability to encapsulate mRNA (Table S2), presumably due to the increased hydrophobicity and self-assembling ability after the attachment of two branch tails."

In supplementary Figure S3, the authors have now reported the physicochemical properties of the LNPs, which is good. A red heat map is used, but it is actually impossible to read the numbers in the dark red fields. In addition, standard deviations and statistics are missing. I suggest that the authors use a table format instead and also report the standard deviation of the numbers. In addition, please also adjust the significant figures. The rule is that the standard deviation provides a measurement of experimental uncertainty and should almost always be rounded to one significant figure. The only exception is when the uncertainty (if written in scientific notation) has a leading digit of 1, then a second digit should be kept. It is clear from the numbers that the mRNA encapsulation efficiency is very low for several of the formulations, and the numbers are in general quite low. It is preferable to have encapsulation efficiencies over 90% to be able to compare different formulations at equal mRNA loading.

Response: We thank the reviewer for these comments. We have now used a table format (Table S2), reported the standard deviation of the numbers and adjusted the significant figures according to the reviewer's suggestions.

We agree with the reviewer that it is preferable to have encapsulation efficiencies over 90% to be able to compare different formulations at equal mRNA loading. However, since these LNPs were formulated by pipette mixing and some lipidoids were naturally inefficient to encapsulate mRNA, we generally obtained mRNA encapsulation efficiencies between 40-80%. MC3 LNP formulated by pipette mixing had similar mRNA encapsulation efficiency (~73.5%) compared to our lead LNPs (**Table S2**). It should be noted that pipette mixing is widely used to accelerate the formulation and screening of LNPs (*Nature Biotechnology*, 2019, 37(10), 1174-1185; *Nature Materials*, 2021, 20(5), 701-710). We have showed that our lead lipidoid obtained from this screening method could achieve >90% mRNA encapsulation efficiency using microfluidic mixing (**Figure 4d**).

When the encapsulation efficiency is low, more lipid is required to achieve the same mRNA dose, and that results in more toxicity. Eventually, high toxicity can cause shut down of protein synthesis. Therefore, the data presented in Figure S4 is of little use, because the formulations are compared at very different lipid doses and at only one single mRNA concentration. Cell viability is determined at different doses, and the concentrations that results in 50% viability should be compared.

Response: We thank the reviewer for these comments. Since we need to compare the mRNA transfection efficiency of our LNPs, we chose to use the same mRNA concentration to treat cells. In this case, lipid doses could be different as the reviewer said. We have now determined cell viability at higher doses and determined the dose that results in 50% viability (IC_{50}). We observed toxicity (< 80% cell viability) for some LNPs at doses of 50 ng/well and 200 ng/well. Most LNPs showed an IC_{50} above 200 ng/well except 1-6 LNP, 1-8 LNP and 1-14-14 LNP.

To avoid the toxicity of lipids and shutdown of protein synthesis, we chose to treat cells at a low mRNA dose (*i.e.*, 15 ng mRNA/well). The purpose for testing cell viability in our study was to confirm that these LNPs were non-toxic at a low mRNA dose, so we could fairly compare their mRNA transfection efficiencies. We have now added more discussion (Page 5, line 167) as follows: “To be noted, we used low doses of mRNA for initial screening to avoid the toxicity of LNPs that could potentially affect protein synthesis (**Figure S3**).”

a**b**

LNP	IC ₅₀ (ng/well)
1-6	148.9
1-8	148.3
1-10	>200
1-12	>200
1-14	>200
1-6-6	>200
1-6-8	>200
1-6-10	>200
1-6-12	>200
1-6-14	>200
1-8-6	>200
1-8-8	>200
1-8-10	>200
1-8-12	>200
1-8-14	>200
1-10-6	>200
1-10-8	>200
1-10-10	>200
1-10-12	>200
1-10-14	>200
1-12-6	>200
1-12-8	>200
1-12-10	>200
1-12-12	>200
1-12-14	>200
1-14-6	>200
1-14-8	>200
1-14-10	>200

1-14-12	>200
1-14-14	167.3
MC3	>200

Figure S3. Cell viability and IC₅₀ of DB-LNPs in Library 1 and other LNPs. (a) Cell viability. HepG2 cells were treated with LNPs at 15 ng mRNA/well (0.24 nM), 50 ng mRNA/well (0.8 nM) or 200 ng mRNA/well (3.2 nM) for 24 h. No obvious cytotoxicity was observed for all LNPs at a low dose (*i.e.*, 15 ng mRNA/well). The dashed line indicates 80% cell viability. Data are presented as mean \pm SD (n = 3). (b) Half-maximal inhibitory concentration (IC₅₀). IC₅₀ was determined by nonlinear regression of dose and cell viability using GraphPad Prism 10.

There is a mistake in the legend of Figure S3: The zeta potential is not determined by using dynamic light scattering, but by laser-Doppler electrophoresis. It is of little use to measure and compare the hydrodynamic size and the zetapotential in serum-containing medium by using the Zetasizer Nano. Both types of measurements are very dependent on the composition of the medium, and the presence of serum proteins interferes with the measurements. Other methods should be used to measure size and zeta potential in serum-containing medium, so I would suggest to take the data for the measurements in FBS-containing PBS out of the manuscript.

Response: We thank the reviewer for the correction and the suggestion. We have now turned previous **Figure S3** into **Table S2** and corrected this mistake. We have now removed the data for the measurements in FBS-containing PBS and removed their corresponding discussion in the manuscript.

It was apparently not clear why it is preferable to provide the RNA concentration on a molar basis instead of a weight basis. The reason why RNA concentrations should be converted to molar concentration is to be able to compare the efficacy and safety of LNPs loaded with RNAs with different lengths, also between different studies. If you use a weight-based concentration, you can only perform a direct comparison of concentrations of RNAs with exactly the same length.

Response: We thank the reviewer for these comments. We have now provided molar concentration in the legends of **Figures 2b, 3b, S3a and S8**. The reason for researchers in both academia and industry prefer to use weight-based concentration instead of molar concentration could be that mRNA is a macromolecule and its molecular weight is variable based on its chemical modification and composition (*e.g.*, 5' cap and the 3' poly(A) tail). If molar concentration is used, this value is typically estimated. For example, in our case, HepG2 cells were treated with LNPs at a dose of 15 ng mRNA/well, which is equivalent to 0.24 nM. We calculated it based on the length of our Luciferase mRNA (1929 nucleotides), use of 1-methyl pseudouridine,

and mean molar mass of 330 Da per nucleotide. Therefore, it is more convenient to use weight-based concentration, which can be easily determined by measuring OD₂₆₀ or Quant-iT RiboGreen RNA assay.

Specific comments:

Page 2, line 56: i.e. is written in italic

Response: We thank the reviewer for this correction. We have now corrected it and others throughout the manuscript based on the reviewer's suggestion.

Page 2, line 57: Please call the COVID-19 mRNA vaccines by their names: Spikevax® and Comirnaty®

Response: We thank the reviewer for this suggestion. We have now revised it according to the reviewer's suggestion.

Page 2, line 58: “Instable” should be corrected to “unstable”

Response: We thank the reviewer for this correction. We have now corrected it based on the reviewer's suggestion.

Page 3, line 96: “..LNP that was..” should be “..LNPs that were” or “...LNP formulation that was...”

Response: We thank the reviewer for this correction. We have now corrected it based on the reviewer's suggestion as follows: “Multiple DB-lipidoids were identified to form potent LNPs for *in vivo* mRNA delivery, which were comparable to or more efficient than the benchmark DLin-MC3-DMA (MC3) LNP formulation that was approved for hepatic delivery of small interference RNA (siRNA).”

Page 12, lines 372-374: Please adjust the significant figures. Again, the rule is that the standard deviation provides a measurement of experimental uncertainty and should almost always be rounded to one significant figure. The only exception is when the uncertainty (if written in scientific notation) has a leading digit of 1, then a second digit should be kept. Hence, 58770 ± 1087 pg/mL should be 58.8 ± 1.1 ng/mL, and 11764 ± 1064 pg/mL should be 11.8 ± 1.1 ng/mL.

Response: We thank the reviewer for this correction. We have now corrected them based on the reviewer's suggestion.

Table S5: A table with only two numbers does not make sense. These two numbers can be written in the text or added to figure 6. Again, please adjust the

significant figures as above.

Response: We thank the reviewer for this suggestion. We have now removed previous Table S5, added these data into Figure 6a and adjusted the significant figures.

Figure 6. DB-LNP-mediated hepatic delivery of FGF21 mRNA. (a) LNP-mediated FGF21 mRNA delivery ($n = 4$). Male mice were fed a HFD for three weeks to induce obesity (body weight ~ 30 g) and fatty liver. These obese mice were i.v. injected with FGF21 mRNA-loaded LNPs at an mRNA dose of 0.25 mg/kg. Serum was collected at the indicated time points for ELISA analysis of FGF21. AUC of FGF21 exposure during the time interval 0–48 h was determined. (b) A scheme of FGF21 mRNA therapy in HFD-induced obese mice. Obese mice were i.v. injected with various LNP formulations at an mRNA dose of 0.25 mg/kg every other day for three doses. Male mice fed a normal chow diet (NCD) was used a control group. (c) Body weight growth curve ($n = 4$). (d) Body weight on Day 6 ($n = 4$). (e) Liver weight ($n = 4$). (f) Representative images of liver histological examinations (H&E staining). Scale bars: 100 μ m. (g) Liver steatosis score ($n = 12$). A total of 12 images for each group (three random fields for each liver section, four mice per group) were analyzed semi-quantitatively for liver steatosis. Data are presented as mean \pm SD.

Reviewer #3 (Remarks to the Author):

I co-reviewed this manuscript with one of the reviewers who provided the listed reports as part of the Nature Communications initiative to facilitate training in peer review and appropriate recognition for co-reviewers.

Response: We thank the reviewer for co-reviewing this manuscript, and hope that the reviewer enjoys our revised manuscript.

REVIEWERS' COMMENTS

Reviewer #2 (Remarks to the Author):

The authors have addressed all of my concerns. I am happy with the improved revised version of this manuscript.

Reviewer #3 (Remarks to the Author):

I have co-reviewed the manuscript with one of the reviewers, and thus my comments are combined in the reviewer's report.

REVIEWERS' COMMENTS AND AUTHORS' ANSWERS

Note: Our responses (standard typeface) to reviewers' comments (bold); the yellow highlighted words and sentences have been added to the main text.

Reviewer #2 (Remarks to the Author):

The authors have addressed all of my concerns. I am happy with the improved revised version of this manuscript.

Response: We thank the reviewer for this encouraging decision. We thank the reviewer again for their time and effort in helping us improve the manuscript.

Reviewer #3 (Remarks to the Author):

I have co-reviewed the manuscript with one of the reviewers, and thus my comments are combined in the reviewer's report.

Response: We thank the reviewer for this encouraging decision. We thank the reviewer again for co-reviewing this manuscript.